# Reprogrammed tracrRNAs enable repurposing of RNAs as crRNAs and sequence-specific RNA biosensors

Yang Liu[1,2], Filipe Pinto [2,8], Xinyi Wan [2,8], Zhugen Yang [3,4], Shuguang Peng[5], Mengxi Li[2], Jonathan M. Cooper [6], Zhen Xie [5], Christopher E. French[2,7] & Baojun Wang [1,2,3,7 ✉]

In type II CRISPR systems, the guide RNA (gRNA) comprises a CRISPR RNA (crRNA) and a hybridized trans-acting CRISPR RNA (tracrRNA), both being essential in guided DNA targeting functions. Although tracrRNAs are diverse in sequence and structure across type II CRISPR systems, the programmability of crRNA-tracrRNA hybridization for Cas9 is not fully understood. Here, we reveal the programmability of crRNA-tracrRNA hybridization for *Streptococcus pyogenes* Cas9, and in doing so, redefine the capabilities of Cas9 proteins and the sources of crRNAs, providing new biosensing applications for type II CRISPR systems. By reprogramming the crRNA-tracrRNA hybridized sequence, we show that engineered crRNA-tracrRNA interactions can not only enable the design of orthogonal cellular computing devices but also facilitate the hijacking of endogenous small RNAs/mRNAs as crRNAs. We subsequently describe how these re-engineered gRNA pairings can be implemented as RNA sensors, capable of monitoring the transcriptional activity of various environment-responsive genomic genes, or detecting SARS-CoV-2 RNA in vitro, as an Atypical gRNA-activated Transcription Halting Alarm (AGATHA) biosensor.

[1] College of Chemical and Biological Engineering & Hangzhou Innovation Center, Zhejiang University, Hangzhou 311200, China. [2] Centre for Synthetic and Systems Biology, School of Biological Sciences, University of Edinburgh, Edinburgh EH9 3FF, UK. [3] Research Centre for Biological Computation, Zhejiang Laboratory, Hangzhou 311100, China. [4] Cranfield Water Science Institute, School of Water, Environment and Energy, Cranfield University, Cranfield MK43 0AL, UK. [5] Center for Synthetic and System Biology, Department of Automation, Beijing National Research Centre for Information Science and Technology, Tsinghua University, Beijing 100084, China. [6] Division of Biomedical Engineering, James Watt School of Engineering, University of Glasgow, Glasgow G12 8QQ, UK. [7] Zhejiang University-University of Edinburgh Joint Research Centre for Engineering Biology, Zhejiang University International Campus, Haining 314400, China. [8] These authors contributed equally: Filipe Pinto, Xinyi Wan. ✉email: baojun.wang@zju.edu.cn

The type II clustered regularly interspaced short palindromic repeats (CRISPR) system employs a small non-coding RNA known as trans-activating CRISPR RNA (tracrRNA) to form the mature dual-RNA structure of the guide RNA (gRNA)[1–5]. By complementary pairing with a precursor CRISPR RNA (pre-crRNA), the tracrRNA mediates RNase III-dependent RNA processing to generate mature crRNA and Cas9 ribonucleoprotein (RNP) complex[3–8]. The tracrRNA thus plays a vital role in the maturation of gRNA for the CRISPR function and has molecular interactions with Cas9 protein[4,9,10].

Several previous studies support the potential programmability of the crRNA-tracrRNA pairings. For example, it has been shown that the Cas9 protein of specific bacteria can form a functional complex with tracrRNAs, having similar secondary structures but different sequences[8,11]. Alternatively, the tetraloop stem of the sgRNA, equivalent to the crRNA-tracrRNA pairing fragment, has been programmed for the *Streptococcus pyogenes* Cas9 (SpCas9)[12,13]. Finally, it has been demonstrated that the CRISPR/Cas9 system can tolerate single base-pair substitutions in the crRNA-tracrRNA pairing region[14]. Despite this knowledge, only during the preparation of this manuscript, the programmability of the crRNA-tracrRNA hybridization region was reported as a mechanism to detect RNA molecules[15].

Here, we now systematically explore the programmability of the crRNA-tracrRNA pairing in the CRISPR/SpCas9 system, with the intention of developing programmable AND logic devices. We validate the programmability of crRNA-tracrRNA pairing and reveal a set of principles to design CRISPR systems with reprogrammed dual-RNA interactions. The high programmability of crRNA-tracrRNA pairing also brings new perspectives and potential applications for the engineering and application of the CRISPR/Cas9 tool, enabling orthogonal AND logic gates to be developed based upon this mechanism.

Further, we show that by reprogramming the crRNA-tracrRNA pairing, SpCas9 can specifically repurpose various RNAs as crRNAs to trigger CRISPR functions. Using these principles, we develop an RNA sensor able to hijack endogenous RNA molecules as crRNAs. Notably, we successfully monitor the transcription level of endogenous genes in *Escherichia coli* and connect the bacterial endogenous genetic network to an artificial gene circuit, which works as a programmable whole-cell biosensor. Additionally, this strategy results in a RNA sensory technique, named AGATHA, which we demonstrate by targeting the detection of SARS-CoV-2 RNA in vitro.

Our study redefines the application capabilities of Cas9 proteins and the sources of crRNAs, providing scope for further studying type II CRISPR systems.

## Results

**crRNA-tracrRNA hybridization is programmable and enables design of orthogonal AND gates.** To study crRNA-tracrRNA pairing, we designed a crRNA-tracrRNA mediated CRISPR activation (CRISPRa) device in bacteria, adapted from our previously reported eukaryote-like CRISPRa system[16,17]. A crRNA-tracrRNA mediated CRISPR activation device requires splitting the sgRNA into crRNA and tracrRNA. Thus, we redesigned the sgRNA by moving the RNA aptamers to the 3'-end of the sgRNA, allowing the splitting of the sgRNA without destroying the RNA aptamers (Supplementary Fig. 1a, b). Next, we split the sgRNA into tracrRNA and crRNA at the tetraloop position (Fig. 1a). An experiment permuting three induction conditions indicated that only when crRNA, tracrRNA and nuclease-dead Cas9 (dCas9) were all induced did the CRISPRa system give the highest output (Fig. 1b, c). The activation efficiencies of crRNA-tracrRNA

mediated CRISPRa and sgRNA mediated CRISPRa are all of the same order of magnitude (Fig. 1c).

We subsequently used the above crRNA-tracrRNA mediated CRISPRa device as a platform to study the programmability of crRNA-tracrRNA pairings. Firstly, we reprogrammed the base pairs of the core crRNA-tracrRNA hybridizing region[11], targeting the core matching region which includes nine RNA base pairs and two wobble base pairs (G-U). Two bases, GA, in crRNA and four bases, AAGU, in tracrRNA form a bulge structure, which is essential for CRISPR/Cas9 functions[12] (Fig. 1d). A sequence of mutations was introduced into the crRNA with complementary mutations made in tracrRNA for testing (Fig. 1d). The results show that all of the outputs were of the same order of magnitude as that of the WT crRNA-tracrRNA matching (Supplementary Fig. 1c). We then explored the orthogonality of mutated crRNA-tracrRNA pairs by recombining the mutated crRNAs and tracrRNAs. Interestingly, the upstream region near the bulge is less able to tolerate mismatches than the region immediately downstream (Fig. 1e). For SpdCas9, more than two adjacent mismatches in the upstream region of the bulge is sufficient to inhibit the CRISPRa function.

We also investigated the dynamic behavior of the dual-RNA mediated CRISPRa system in order to show the Boolean logic profile of an AND gate[18]. Although AND-gate circuits have various applications in synthetic biology[19], building complex artificial genetic networks requires a large number of highly orthogonal AND gates. We subsequently showed that CRISPRa based on the programmability of crRNA-tracrRNA pairing can efficiently address this problem (Fig. 1f).

Since inducible systems are preferred as a mechanism to control logic gate devices[16], we used the induced or non-induced states of the crRNA or tracrRNA to represent states 1 and 0, respectively. For the inducible system, this AND gate became less efficient due to its high sensitivity to tracrRNA leakiness from the inducible promoter input (Fig. 1c). We truncated the crRNA-tracrRNA pairing region to 14 bp to weaken their pairing affinity and improved the AND gates with good symmetry in response to the two inputs (Fig. 1g). In addition, inspired by our previous experience with dxCas9 3.7, and how it could be used to optimize the sensitivity to the sgRNA leakiness of our CRISPRa system[16], we verified that dxCas9 can also work with our newly designed sgRNA and performed better than dCas9 (Supplementary Fig. 2). Next, we showed that the dxCas9 could make the AND gates more symmetrical in response to the two inputs (Supplementary Fig. 3).

We randomly generated four 14 bp (12–13 bp for non-WT sequences, since there may not be wobble base pairs) sequences for the crRNA-tracrRNA matching region and tested their orthogonality and functionality within the AND gates. Accordingly, we showed that orthogonality could be achieved following the principle of complementary base pairing, and that all circuits retain their function as AND gates (Fig. 1h, i).

**CRISPR activity with reprogrammed crRNA-tracrRNA pairings depends upon multiple factors.** For any engineering purpose based on reprogrammed tracrRNA-mediated CRISPR, the structural or sequence preference of SpCas9 for the crRNA-tracrRNA matching region is an important guiding principle. Based upon this, we first investigated the structural preference by varying the length of the RNA hybridization segment. We truncated the WT crRNA and WT tracrRNA simultaneously (Fig. 2a), and noticed a dramatic decrease of the CRISPRa output when the paired length is shorter than 14 bp (including the two wobble base pairs). In contrast, the 14 bp length could support a similar output level to that from the WT version (Fig. 2a, Supplementary Fig. 4).

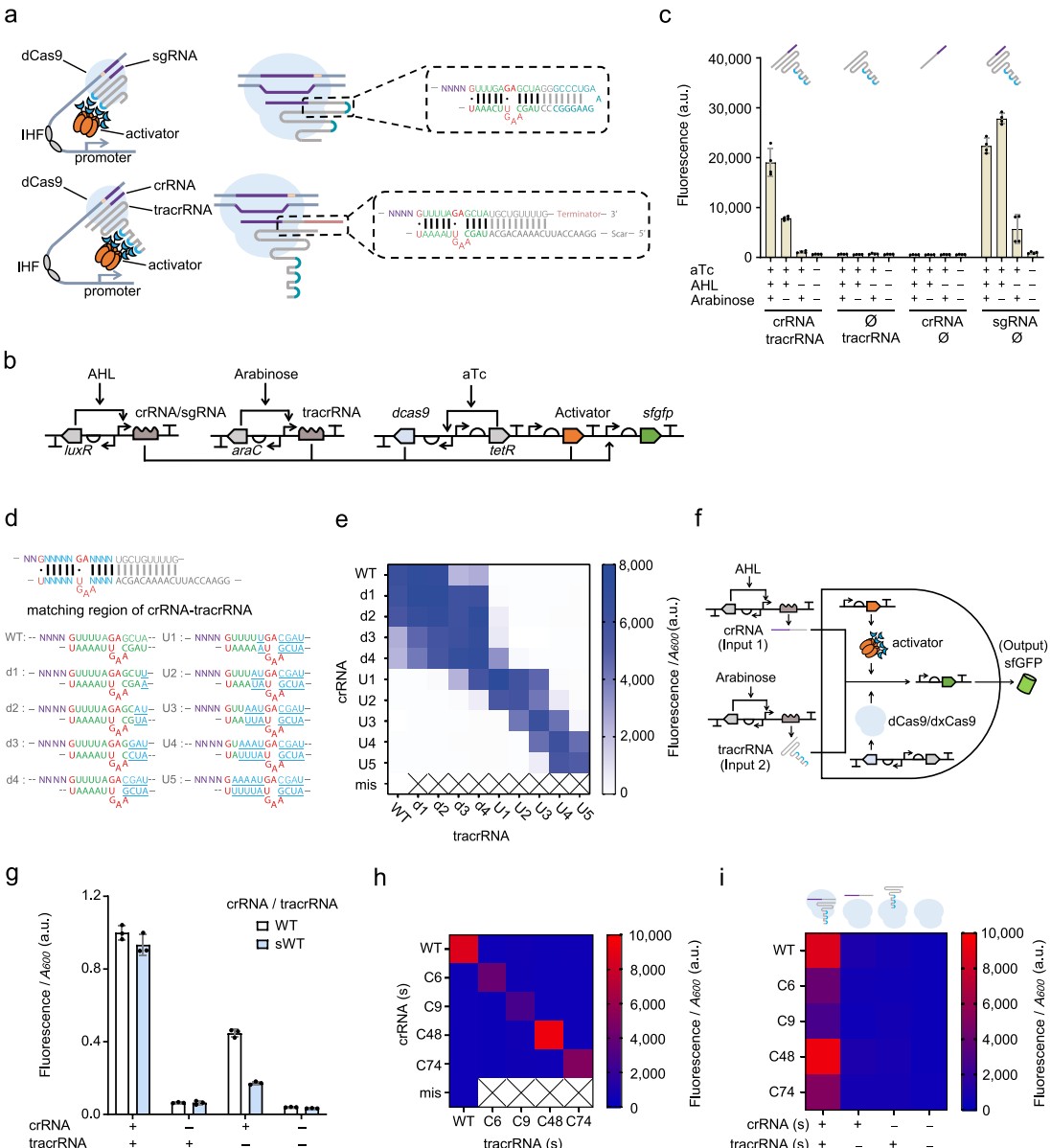

**Fig. 1 Dual-RNA mediated CRISPRa system reveals the programmability of crRNA-tracrRNA hybridization. a** Schematics showing the original sgRNA mediated eukaryote-like CRISPRa system (top) and the dual-RNA mediated CRISPRa system (bottom). Green base pairs, the minimum structure required for CRISPR/Cas9 function; blue bases, RNA aptamer BoxB; red bases, the bulge structure and wobble base pairs (marked as black dots). **b** Circuit design of the crRNA-tracrRNA mediated CRISPRa system. The activator PspFΔHTH::λN22plus is driven by constitutive promoter J23106[38]. The anhydrotetracycline (aTc), N-(3-oxohexanoyl)-L-homoserine lactone (AHL), and arabinose are the inducers for promoters $P_{tet}$, $P_{lux2}$, $P_{BAD}$, respectively. **c** Test of the combination of three components in the crRNA-tracrRNA mediated CRISPRa, and comparison of the function of gRNAs before and after being split. Induction combinations were achieved via presence (+) or absence (−) of inducers. Error bars, mean values +/− s.d. ($n = 4$). **d** Design of the library for testing reprogrammed crRNA-tracrRNA hybridized pairs. The blue letter 'N' indicates the substituted base pairs. The representation of motifs in the remaining sequences on the bottom is the same as those in **a**. **e** Orthogonality test of reprogrammed crRNA-tracrRNA pairs. The label 'mis' indicates the WT crRNA has a mismatched spacer (LEB3) with the target UAS (LEA2). The reprogrammed tracrRNAs and crRNAs were combined in pairs. ($n = 3$). **f** Design of orthogonal AND gates based on reprogrammed crRNA-tracrRNA paring. **g** The AND gate function based on the original tracrRNA design and the truncated tracrRNA. Normalization was performed with the output of WT tracrRNA-based device at 'On' state. Error bars, mean values +/− s.d. ($n = 3$). **h** The heat map shows the outcome of an orthogonality test of the WT and 4 randomly generated crRNA-tracrRNA paired sequences. The '(s)' indicates that the short 14 bp version of crRNA-tracrRNA pairs was used in this test. ($n = 3$). **i** Each orthogonal crRNA-tracrRNA mediated CRISPRa device displays the Boolean AND logic profile. The cartoon above shows the presence or absence of the components of the CRISPR complex under different induction conditions. The data in the active state and induction conditions are the same as in **h**. ($n = 3$). a.u., arbitrary units. Source data are provided as a Source Data file.

We also investigated whether RNase III was necessary for dual-RNA-mediated CRISPRa through engineered designs where both the crRNA and tracrRNA have ends longer than mature WT dual-RNA processed by RNase III. In previous studies, RNase III was shown to be indispensable for the maturation of crRNA and the immune function of Cas9[6]. To determine whether RNase III was required in our current study, we introduced our dual-RNA-mediated CRISPRa device into *E. coli* strains W3110 and its

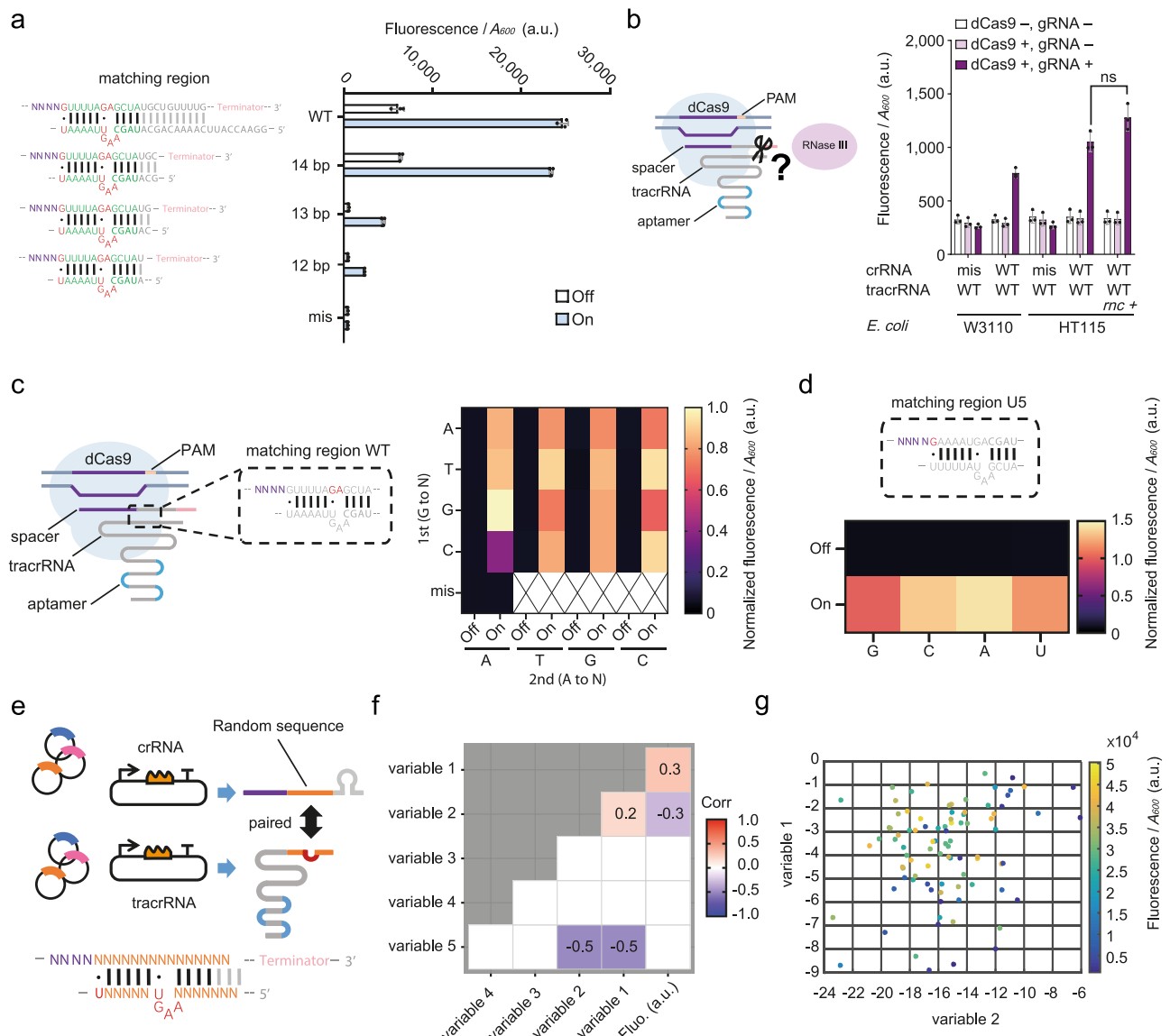

**Fig. 2 Mode of action of reprogrammed crRNA-tracrRNA pairs. a** The length of the hybridized region affects the CRISPR function. The diagram on the left depicts the sequences used and the bar chart on the right exhibits the CRISPRa output. The label 'mis' indicates the WT crRNA has a mismatched spacer (LEB3) with the target UAS (LEA2). Error bars, mean values $+/-$ s.d. ($n = 3$). **b** RNase III is unnecessary for dual-RNA mediated CRISPRa. The label 'mis' is the same as in **a**. Statistical difference was determined by a two-tailed $t$ test: $p = 0.0615$, $t = 2.577$. Error bars, mean values $+/-$ s.d. ($n = 3$). **c** Saturated mutations of the GA site. The diagram on the left shows location of the GA site. The heat map on the right shows the results. All the genes were induced for 'On' state, no induction for the 'Off' state. Data normalized by the output level from the 'GA' at 'On' state. ($n = 4$). **d** Saturated mutations of the base guanine in the core hybridized region. The U5 crRNA-tracrRNA pair was used as an original crRNA version, which had been shown as a functional crRNA-tracrRNA pair (see in Fig. 1e). The data was normalized by the output level from the sample with the WT U5 'G' at 'On' state. ($n = 4$). **e** Schematic showing the randomized sequence library of crRNA-tracrRNA pairing. The orange bases show the paired randomized sequences, and purple 'N's stand for the spacer region, a fixed sequence (LEA2) among all variants. The red bases in tracrRNA are not changed. **f** Pearson correlation coefficient between individual features (variables, see METHODS) and the output fluorescence level. Blank indicates no significant correlation ($p > 0.05$). Numbers represent correlation coefficients (Corr). **g** Reprogrammed tracrRNA-crRNA library represented by variable 2 for crRNA-tracrRNA heterodimer binding and variable 1 for crRNA folding. Each dot represents a sample from the library. Samples are colored according to their average fluorescence level ($n = 3$); a.u., arbitrary units. $p$ value summary: ****$p$ value < 0.0001, 0.0001 < ***$p$ value < 0.001, 0.001 < **$p$ value < 0.01, 0.01 < *$p$ value < 0.05, $p$ value $\geq$ 0.05: ns. Source data are provided as a Source Data file.

RNase III gene (*rnc*) knockout strain HT115. Our results showed that RNase III is not required for our CRISPRa system. Furthermore, we complemented strain HT115 with a copy of the *rnc* gene expressed from a plasmid. However, this did not significantly improve the function of CRISPRa (Fig. 2b).

Subsequently, we investigated the importance of the three bases in the wobble base pairs (G - U) and the bulge in crRNA. By introducing saturation mutation at the GA site of the WT hybridization sequence,

and at the first wobble base pair (G - U) of a functional hybridization sequence U5, we confirmed that changing these compositions did not completely disrupt CRISPRa function, although we observed that a small number of mutations can result in changes in CRISPR activity. Importantly, our results confirmed that the base pairing at a specific position does not determine the function of CRISPR alone (Fig. 2c, d).

The results above provide a conveniently simplified crRNA-tracrRNA model, providing the ground rules to enable us to build a

library with 90 paired candidates, including random sequences in the crRNA-tracrRNA matching region (Fig. 2e, Supplementary Data 1). As a boundary condition in the library design, we ruled out the presence of NGG immediately downstream of the spacer, in order to prevent a CRISPRi effect on the crRNA generator. Interestingly, about 73% of candidates show higher than 50 % of the activity of the WT sequence, and about 40 % of candidates even show higher activity than the WT sequence (Supplementary Fig. 5).

Subsequently, in order to explore potential factors influencing the activity of these reprogrammed tracrRNA-crRNA pairs, we calculated the Pearson correlation coefficients between six selected features measured as an output fluorescence. Figure 2f shows that only the variables reflecting crRNA secondary structure (variable 1) and crRNA-tracrRNA affinity (variable 2) have a significant correlation with the fluorescence reporter signal ($p < 0.05$). We were thus able to build a linear regression model using these two variables to reveal their relationship. Despite the weak correlation, the available data provides a trend, consistent with our understanding of the CRISPR/Cas9 complex which is promoted by high affinity between crRNA and tracrRNA and weak secondary structure of crRNA itself (Fig. 2g).

**Reprogrammed tracrRNA is able to repurpose mRNA as crRNA**. This ability to program the crRNA-tracrRNA pairing, demonstrated above, indicates the exciting potential of hijacking of various RNAs as crRNAs. The natural crRNA mainly includes two sequence motifs, namely the spacer, matched with target DNA and the downstream repeat matched with tracrRNA. Although pre-crRNA processing is necessary for CRISPR function[6,20], the engineering of sgRNA proves that the 3'-end of crRNA near Cas9 is unnecessary for Cas9 function[4]. In addition, the extended 5'-end of sgRNA and extended 3'-end of crRNA does not disrupt the activity of CRISPR/Cas9[21–23].

By combining design constraints around the programmability of the spacer sequence and its downstream repeat region, we can infer that the sequence of the whole crRNA can be altered without destroying the function of the CRISPR/Cas9 system. Subsequently, we deduced that any RNA may become a crRNA through dual recognition by a matched programmed tracrRNA and target DNA.

In order to test this hypothesis, we randomly selected three GA sites (R1, R2 and R3) on the mRNA encoding red fluorescent protein (RFP), since GA provided the highest function in our saturation mutation test (Fig. 2c). The presence of NGG adjacent to the assumed spacers was deliberately avoided. According to the context sequence adjacent to the GA sites, we designed three corresponding tracrRNAs and three cognate promoters. Each of the promoters had the upstream activating sequence (UAS) matching the assumed spacer, which was the upstream mRNA region immediately adjacent the predicted mRNA-tracrRNA hybridization position (Fig. 3a, Supplementary Data 1).

Our result showed that, when employing mRNA-paired tracrRNAs, there was a positive linear correlation between the expression level of RFP and GFP for the R2 and R3 sites. This correlation disappears when using the mis-matched WT tracrRNA (Fig. 3b, c). The three randomly selected tracrRNA target sites gave different CRISPRa efficiencies (Fig. 3b). For all the sites, when using the corresponding sequences in isolation as a fragmented RNA, the CRISPRa output intensities obtained were greater than using the whole mRNA (Fig. 3d, Supplementary Fig. 6). We also noted that the relationship of CRISPRa strength between different target sites changed dramatically when the target sequences were fragmented short RNAs or different hybridization lengths were employed (Supplementary Fig. 7).

Surprisingly, we noted that the binding of the CRISPR complex to mRNA did not interfere with RFP translation (Fig. 3b), indicating that the translation process does not cause the output discrepancy between mRNA and isolated fragments, described above. Similarly, in order to assess whether the translation process interferes with the CRISPRa system, a version of mRNA lacking the ribosome binding site (RBS) was used to test the impact of translation on CRISPRa. The results showed that although the level of RFP was reduced due to the lack of RBS, the output of mRNA-mediated CRISPRa on R2 and R3 sites did not change significantly (Supplementary Fig. 8).

We subsequently explored a series of design principles based upon these observations, including (i) the use of a simplified tracrRNA (s-tracrRNA) without redundant 5'-end to improve the performance of this mRNA monitor on all the sites (Fig. 3d, Supplementary Fig. 6), and (ii) optimization of the site selection and tracrRNA by independent methods. By comparing the output patterns from the 3 sites when CRISPRa is combined with mRNA or fragmented mRNA, with or without tracrRNA optimization, we found that the patterns based on complete mRNA and fragmented mRNA were different. However, the output patterns based on presence or absence of tracrRNA simplification maintain the similarity when either the mRNA or fragmented mRNA are used (Fig. 3e).

Finally, we combined the design strategies developed with a positive feedback loop (PFB) to optimize the mRNA monitor in vivo (Fig. 3f). For PFB design, an additional promoter was utilized to produce the additional activator, with the dynamic range of the response subsequently increasing from 9.6-fold in the original device to 52.2-fold with PFB design through the above engineering methods (Fig. 3f, g, Supplementary Fig. 6).

**Monitoring endogenous RNAs via reprogrammed tracrRNA-mediated CRISPRa**. Successful hijacking of the RFP mRNA as crRNA raises an interesting question as to whether the same strategy is available for endogenous RNAs transcribed from the genome? To explore this possibility, we firstly chose the *ars* operon of *E. coli* as a target, since the promoter P$_{arsR}$ can be induced by arsenic, a toxic environmental pollutant[24,25].

Having previously shown that different sites on the mRNA may have different availabilities, we tested four candidate sites on the *ars* transcripts on artificial circuits first. The arsenic responsive gene cluster (*arsRBC*) was isolated and cloned into a vector under inducible promoter P$_{lux2}$. The four candidate sites were made into short fragments of mRNA (Fig. 4a).

By testing the availability of these sites by using programmed corresponding tracrRNAs and promoters, we showed that only Ar2 is an available target (Fig. 4b). Yet, for the fragmented mRNA, different availabilities of the mRNA sites were revealed, similar to what we have observed in hijacking RFP mRNA. Again, this implies that the characteristics of mRNA itself can strongly affect the function of mRNA-mediated CRISPR (Fig. 4b, c).

Next, we used the selected Ar2 site to monitor the activity of the *ars* operon on the *E. coli* genome. The reporter circuit and a tracrRNA-Ar2 generator were first transformed into *E. coli* (Fig. 4d), before constitutively inducing the expression of dCas9, tracrRNA, and activator. When a concentration gradient of sodium arsenite (NaAsO$_2$) was applied, as expected, for tracrRNA-Ar2, an output increase was detected with increasing arsenite concentration (Fig. 4e). In this case, the output signal is weaker than that from the artificially expressed mRNA, which may be due to the relatively low concentration of the endogenous RNA and the different spatial locations of the transcripts in the cell. Finally, we optimized this monitor by

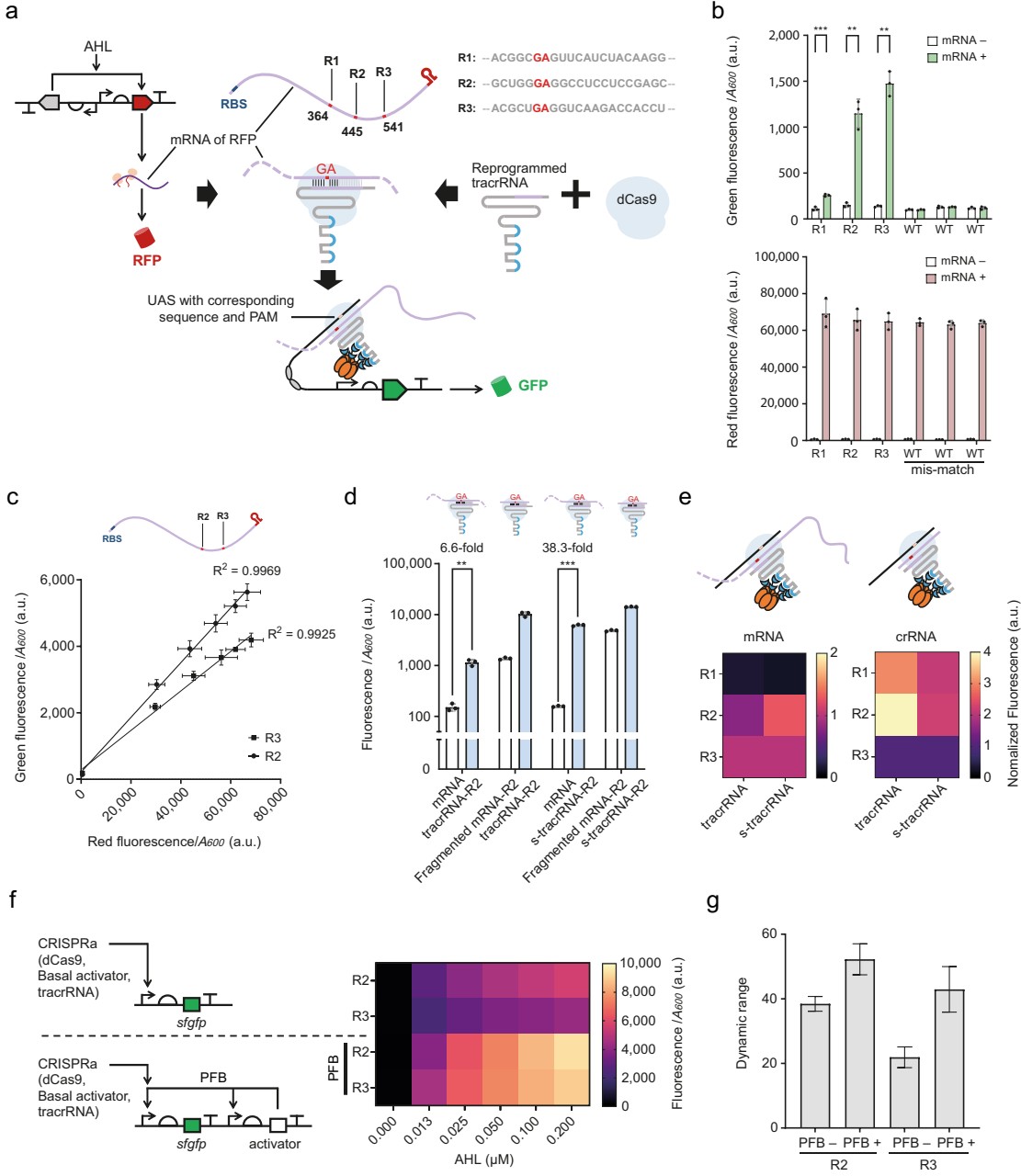

**Fig. 3 Engineered tracrRNAs repurpose mRNAs into crRNAs and activate CRISPRa. a** Schematic showing the design of the mRNA sensor. Three randomly selected RNA sites (red symbol) are represented. The sequences of three sites are shown in grey letters with GA site highlighted in red. **b** Green (top) and red fluorescent outputs (bottom) of the mRNA-mediated CRISPRa. WT tracrRNA was used as a mis-matched control. Statistical difference was determined by a two-tailed $t$ test: R1, $p = 0.0004$, $t = 11.01$, or by a Welch's $t$ test: R2, $p = 0.0070$, $t = 10.91$; R3, $p = 0.0032$, $t = 17.13$. Error bars, mean values ± s.d. ($n = 3$). **c** Scatter plot shows the linear relationship between the expression levels of RFP and sfGFP. For R2, Equation: $GF = 0.08230 \times RF + 216.5$ ($GF$: green fluorescence, $RF$: red fluorescence); the coefficient of determination ($R^2$) is 0.9969; $p$-value <0.0001. For R3, Equation: $GF = 0.05937 \times RF + 282.8$, $R^2 = 0.9925$; $p$-value < 0.0001. Error bars, mean values ± s.d. ($n = 3$). **d** 5′ end truncation on tracrRNA improved CRISPR function. 5′-end extended tracrRNA (tracrRNA) or 5′-end truncated tracrRNA (s-tracrRNA) were employed. For the combination of mRNA and tracrRNA-R2, the same data were used as the data in **b**. Statistical difference was determined by a two-tailed Welch's $t$ test: mRNA + s-tracrRNA-R2, $p = 0.0005$, t = 43.63. Error bars, mean values ± s.d. ($n = 3$). **e** The same sequence from mRNA fragment and whole mRNA resulted in altered CRISPR functions. The raw data obtained from R2 site are the same as that in **d**. In each group, the data are normalized by the CRISPRa output level of the sample with R3 site. **f** Diagram shows the CRISPRa circuit with a positive feedback loop (PFB), with the results on the right. The data of R2 and R3 sites without PFB is same as those in **c**. **g** Dynamic range calculated from **f**. Error bars, mean values ± s.d. ($n = 3$). a.u., arbitrary units. $p$ value summary: ****$p$ value < 0.0001, 0.0001 < ***$p$ value < 0.001, 0.001 < **$p$ value < 0.01, 0.01 < *$p$ value < 0.05, $p$ value ≥ 0.05: ns. Source data are provided as a Source Data file.

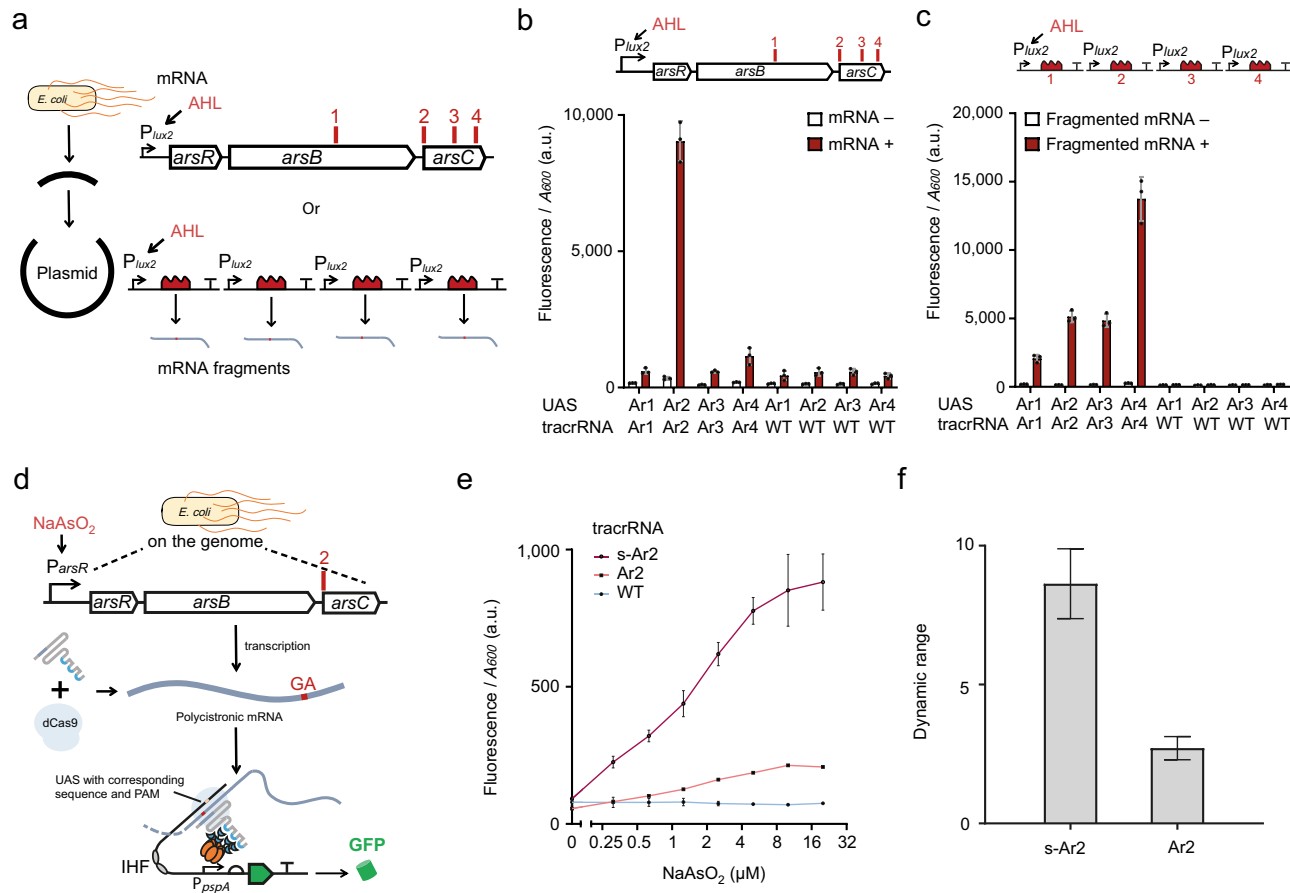

**Fig. 4 Hijacking of endogenous mRNA as crRNA to detect arsenic induced transcription. a** Schematic showing the sites selected for tracrRNA targeting. The DNA encoding the endogenous arsenic responsive pathway in *E. coli* was isolated and inserted into a plasmid under $P_{lux2}$ promoter. In parallel, the four candidate sites were also extracted separately and inserted into the same vector backbone. The red numbers and short lines show positions of the four candidate sites (Ar1-Ar4). **b** Test of the availability of the candidate sites on the plasmid transcribed mRNA. The mRNA expressed by the $P_{lux2}$ promoter was bound to different tracrRNA and corresponding promoters. Error bars, mean values $+/-$ s.d. ($n = 3$). **c** Test of the availability of candidate sites on the fragmented mRNAs as crRNAs. crRNAs expressed by the $P_{lux2}$ promoter were bound to corresponding tracrRNAs and promoters. Inductions are the same as in **b**. Error bars, mean values $+/-$ s.d. ($n = 3$). **d** Schematic showing mechanism of hijacking the endogenous mRNA of the arsenic responsive operon of *E. coli* to activate reporter expression by CRISPRa. Red symbols indicate the Ar2 targeting site in the polycistronic mRNA of the arsenic-related gene cluster. The upstream RNA sequence immediately adjacent to the tracrRNA target site was used as the design template for the UAS for CRISPRa activation (see Supplementary Data 1). **e** CRISPRa outputs of hijacking endogenous mRNA as crRNA. 'Ar2' stands for the sensor with originally designed tracrRNA. The 's-Ar2' stands for the sensor with 5'-end truncated tracrRNA. 'WT' indicates that a WT tracrRNA which cannot match the mRNA used here. A gradient concentration of sodium arsenite (0, 0.31, 0.63, 1.3, 2.5, 5, 10, 20 μM) was used for inducing transcription of the arsenic responsive gene cluster. Error bars, mean values $+/-$ s.d. ($n = 3$). **f** Dynamic range calculated from **e**. Error bars, mean values ± s.d. ($n = 3$). a.u., arbitrary units. Source data are provided as a Source Data file.

simplifying the 5'-end of tracrRNA, and improved its performance (Fig. 4e, f).

We subsequently generalized the same strategy to other environment-responsive genes on the *E. coli* genome, including the oxidative stress-related small RNA OxyS, the mRNA of zinc/lead responsive gene *zraP*, the mRNA of zinc/cadmium responsive gene *zntA*, and the mRNA of copper responsive gene cluster *cusCFAB* were chosen as the targets[26,27].

To do this, we first isolated the four transcriptional units from the *E. coli* genome and inserted them into the AHL-induced circuits. We selected 5 to 9 target sites without any sequence restriction for each target RNA, and then screened the sites using the corresponding tracrRNA generators (with aforementioned optimized design), reporter circuits and AHL induction (Supplementary Fig. 9). By picking the most efficient site on each RNA for further testing the specificity of CRISPRa, we were able to show that the matched tracrRNA is necessary for the CRISPRa function mediated by these transcripts (Supplementary Fig. 10).

Subsequently, we demonstrated that the hijacking of endogenous small RNA OxyS and mRNA of *zraP*, *zntA*, or *cusCFAB* can also be used to trigger CRISPRa for sensing hydrogen peroxide, zinc, lead, cadmium and copper ions, respectively. Among them, we demonstrated that the CRISPRa sensors based on small RNA OxyS, *zraP* mRNA and *cusCFAB* mRNA can detect the presence of hydrogen peroxide, zinc, lead, or copper ions with a high dynamic range. However, the sensor based on *zntA* mRNA has a relatively low dynamic range in response to zinc, indicating the need of further optimization in some specific cases (Supplementary Fig. 11).

Overall, we conclude that endogenous small RNA and mRNAs can be hijacked by dCas9, and used to monitor genomic transcriptional activity and connect the cellular gene regulatory network to an artificial actuating or reporting gene circuit.

**CRISPR/Cas9-based RNA detection in vitro.** Finally, we demonstrated an engineering method that can convert non-

crRNA into crRNA with future potential applications for in vitro nucleic acid sensing. To achieve this, we developed a programmable RNA sensor with unique dual recognition characteristics, which we hypothesized had a higher specificity than the DNA recognition of the CRISPR/Cas9 complex.

Compared with other well-known CRISPR RNA sensors such as SHERLOCK[28], HOLMES[29] and DETECTR[30] that can continuously cleave non-specific single-stranded nucleic acid, Cas9 cleavage lacks a signal amplification effect. In contrast to Cas12a and Cas13, Cas9 can only cleave specific target DNA and then remains bound to it, resulting in insufficient repetitive cleavage of the target DNA[31].

In order to overcome these shortcomings in Cas9-directed RNA sensing, we designed an in vitro transcription-based biosensing reporter named the Atypical gRNA-activated Transcription Halting Alarm (AGATHA) system. It consists of a reporter DNA named AGATHA DNA, fluorogen DFHBI, purified Cas9, and reprogrammed tracrRNA for sensing target RNA in an in vitro T7 expression system[32] (Fig. 5a). The AGATHA DNA can express an inactive Broccoli RNA aptamer[33] which is blocked by a 3'-end secondary structure (anti-Broccoli tail). When sensing the target RNA, the Cas9 is activated and prevents the transcription of the 3' end secondary structure, which will then continuously transcribe a functional aptamer that can bind to DFHBI, leading to an amplified fluorescent output signal (Fig. 5a).

We tested the AGATHA biosensor system using synthetic coronavirus SARS-CoV-2 RNA fragments with corresponding reprogrammed tracrRNAs and Cas9. In this case, we designed a Cas9 cleavage site A between the Broccoli RNA coding sequence and the anti-Broccoli tail in the AGATHA DNA. Only when the SARS-CoV-2 RNA fragment A matched the AGATHA DNA and corresponding tracrRNA A, did AGATHA give a significant report signal (Fig. 5b). Otherwise, even if the AGATHA DNA matched the target RNA, no signal could be detected. This dual recognition mechanism ensures the high specificity of AGATHA detection.

Two different protocols were designed to further test and optimize the AGATHA system (Fig. 5c). Upon sensing the SARS-CoV-2 RNA, we observed fluorescence reported by the AGATHA sensor within 10 min (Fig. 5d). The results also showed that pre-incubation of AGATHA DNA with CRISPR/Cas9 could improve the AGATHA sensor's output dynamic range. After incubation in the plate reader, the Broccoli RNA concentration in the microwells had accumulated sufficiently to be distinguished by the naked eye or by a low-cost cell phone camera (Fig. 5d), demonstrating its potential as a portable rapid diagnostic tool.

## Discussion

In this work, we systematically investigated the programmability of crRNA-tracrRNA pairing in the CRISPR/Cas9 system, based on our previously engineered eukaryote-like CRISPRa device, revealing the tolerance of SpCas9 protein for reprogrammed sequences of the crRNA-tracrRNA hybridization region. We showed that mismatches can confer orthogonality to crRNA-tracrRNA mediated CRISPRa and enable orthogonal AND logic devices.

We studied the structural and sequence preference of SpCas9 to crRNA-tracrRNA hybridization region using a variety of methods including through the development of a random sequence library. It is worth noting that different Cas9 homologs may have different tolerances and preferences to the reprogrammed crRNA-tracrRNA matching sequences and mismatches.

Our previous research had showed dxCas9 exhibited a lower tolerance for continuous mismatches at specific sites in the crRNA-tracrRNA pairing region[16] when compared with dCas9.

Hence, the sequence preference of different Cas9 proteins for the crRNA-tracrRNA matching region may need to be assessed by combining machine learning and high-throughput methods in a case-by-case manner. According to various practical scenarios shown here (i.e., endogenous RNA hijacking or designing AND gates), there are other factors than these sequence preferences which may affect the CRISPRa function. This re-emphasizes the significance of further research on target site availability by combining other methods.

Since CRISPR/Cas9 can tolerate extensions of the crRNA at the 5'-end and 3'-end, with enough length, any RNA may become a crRNA by binding to its complementary tracrRNA. Our experiments confirmed this, demonstrating hijacking of plasmid-transcribed or endogenous small RNA/mRNA molecules as crRNAs to trigger CRISPRa function.

We also demonstrated how reprogrammed tracrRNAs can lead to the cell-free reporter system AGATHA with a signal amplification effect on our RNA sensor based on the programmability of crRNA-tracrRNA hybridization. Our system only amplifies the output signal when it senses target RNA and thus has a low background readout. This system can be used for developing CRISPR/Cas9-mediated RNA sensors and for enhancing or amplifying the signal of other types of CRISPR-based RNA sensors, as well as for many other research and application scenarios related to DNA fragmentation.

These observations are all consistent with a recently published study by Jiao et al.[15] which described the use of noncanonical crRNAs by Cas9. Their work when read with ours, shows that synthetic biology can use engineering logic to provide insights into previously unrecognized mechanisms from natural evolution. In terms of their engineering applications, these revealed capabilities of reprogrammed tracrRNAs, used to develop the SARS-CoV-2 sensor (AGATHA and LEOPARD[15]) demonstrate that different engineering designs and inspirations can lead to congruent and related innovations that are functionally and mechanistically independent of each other[15].

We believe that in future the reprogramming of the CRISPR/Cas9 system, and its use to develop engineered systems able to recognize RNA molecules, either as in vitro or in vivo biosensors, will become an emerging field that has great potential. The tolerance of dCas9 for gRNA sequence diversity also raises an intriguing question as to whether it may be possible for endogenous RNA in eukaryotic cells to coincidentally form a functional CRISPR complex with Cas9, and then deliver an off-target effect. Given that Cas9 binding to non-canonical RNA has recently been reported in bacteria[15], exploration of the interaction between endogenous RNA and Cas9 in eukaryotes may lead to new technologies including sensitive imaging reporters to explore gene activation, or even therapeutics, in mammalian systems.

In summary, we believe that the programmability of crRNA-tracrRNA hybridization and the RNA recognition capability of the CRISPR/Cas9 system opens up a significantly broader picture of CRISPR/Cas9 engineering. For example, this programmable system may be utilized to sense RNA in vivo or in vitro, to build RNA editors, to reconstruct the topology of the genetic regulatory networks, to mark transcripts in situ, or even to build complex cellular computing circuits, all with potential applications in biosensing and therapeutics. Our findings have elevated the current knowledge of CRISPR systems and will enable the expansion of current CRISPR technologies in the future.

## Methods

**Strains and growth conditions**. Unless otherwise specified, all the assays were performed in *E. coli* strain MC1061Δ*pspF*. The *E. coli* MC1061Δ*pspF* was generated through P1 phage transduction, using *E. coli* strain BW25113Δ*pspF*739::*kan* from

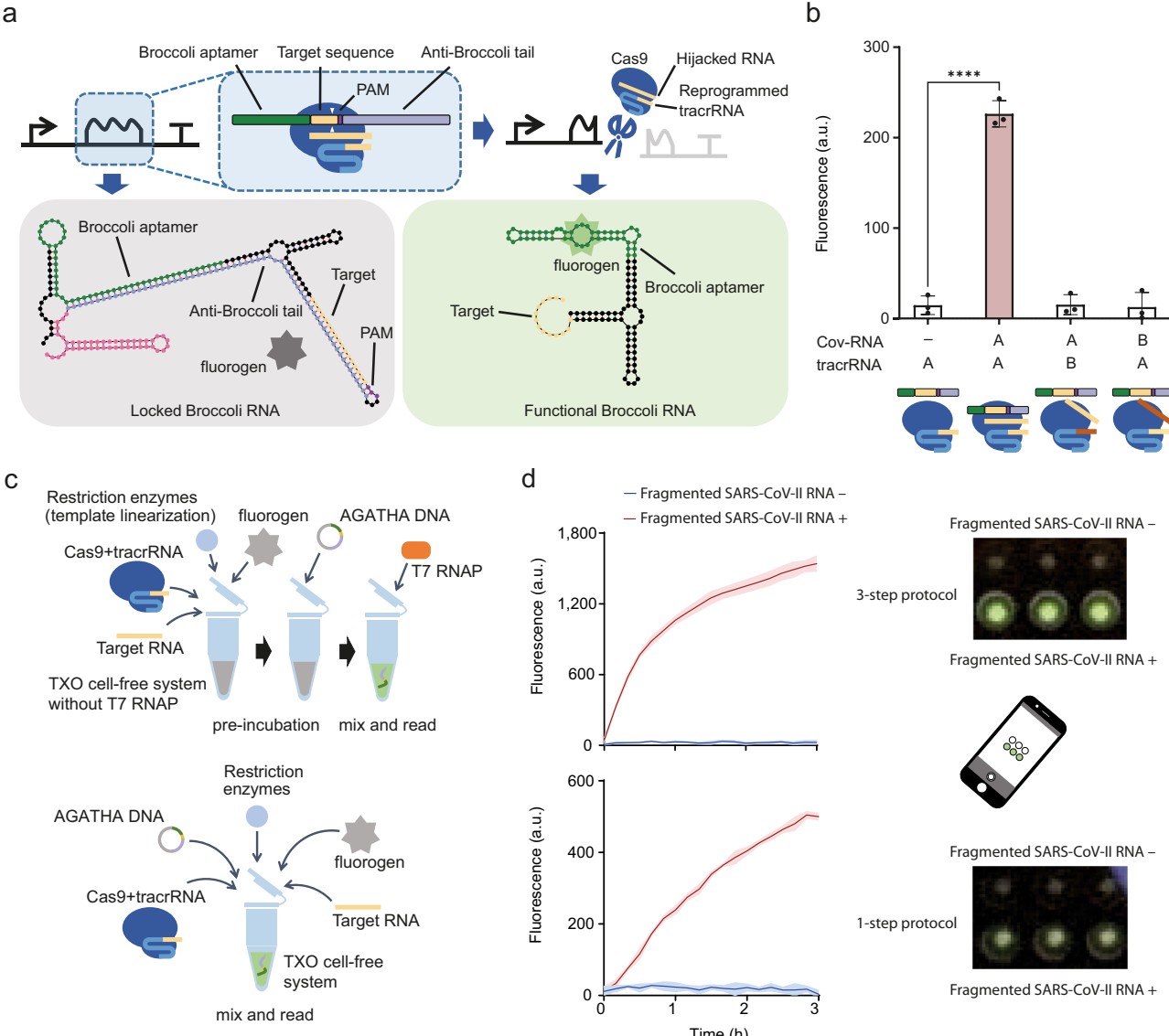

**Fig. 5 Atypical gRNA-activated Transcription Halting Alarm (AGATHA) system. a** Schematic showing the design and mechanism of AGATHA sensor. In the box surrounded by the dashed line, the bordered green bar indicates the Broccoli RNA sequence, and the bordered yellow bar indicates the CRISPR target DNA site paired with the target RNA. The bordered dark purple bar indicates PAM, and the bordered light purple bar indicates an anti-Broccoli tail with an inverted repeat sequence of the Broccoli. The two white triangles mark the position where CRISPR/Cas9 cleaves the AGATHA reporter DNA; the borderless lines indicate gRNA molecules. The cartoon below shows the secondary structure of the RNA aptamer predicted by mfold Web Server[39]. The polygonal star is the fluorogen, which is activated when binding to the Broccoli aptamer. The colors in the RNA structure diagram are consistent to those in the box above, with the pink segment indicating the T7 terminator. **b** Specificity test of AGATHA. Two different viral RNA fragments (A and B) from the SARS-CoV-2 virus and the corresponding reprogrammed tracrRNAs were used to test the specificity of AGATHA. The T7 in vitro transcription system was used on addition of 5 ng μL$^{-1}$ AGATHA DNA, 50 nM synthetic CoV-SARS-2 RNA fragment, 50 nM synthetic tracrRNA and 50 nM SpCas9. Data were collected 3 h following incubation. Statistical difference was determined by a two-tailed $t$ test: R1, $p < 0.0001$, $t = 20.57$. Error bars, mean values ± s.d. ($n = 3$). **c** Schematic showing the assay procedure of AGATHA to detect target coronavirus (SARS-CoV-2) RNA fragments. The reaction was carried out either with (top) or without (bottom) pre-incubation (see METHODS). **d** Dynamic output responses (left) and mobile phone images (right) of the AGATHA sensor responding to SARS-CoV-2 RNA fragment with (top) or without (bottom) pre-incubation. Phone images were acquired at the end of the reactions (see METHODS). The T7 in vitro transcription system is the same as in **b**. Error bars, mean values ±s.d. ($n = 3$). a.u., arbitrary units. $p$ value summary: ****$p$ value < 0.0001, 0.0001 < ***$p$ value < 0.001, 0.001 < **$p$ value < 0.01, 0.01 < *$p$ value < 0.05, $p$ value ≥ 0.05: ns. Source data are provided as a Source Data file.

the Keio collection as the donor strain[16,34]. For circuit construction, both *E. coli* TOP10 and *E. coli* DH5α were used in this study.

For bacterial culture, we cultured *E. coli* in Lennox's Lysogeny Broth (LB-Lennox) medium (10 g L$^{-1}$ peptone (EMD Millipore), 5 g L$^{-1}$ yeast extract (EMD Millipore), 5 g L$^{-1}$ NaCl (Fisher Scientific)) with appropriate antibiotics for all the experiments. Antibiotics were applied at final concentrations of: 50 μg mL$^{-1}$ ampicillin (Sigma-Aldrich), 25 μg mL$^{-1}$ kanamycin (Sigma-Aldrich), and 12.5 μg mL$^{-1}$ chloramphenicol (Sigma-Aldrich)[16].

Before gene expression assays, unless otherwise stated, we co-transformed the desired circuits into chemically competent *E. coli* MC1061Δ*pspF* cells using a typical chemical transformation protocol (i.e. adding 1 μL of each plasmid DNA with concentration from 10–100 ng μL$^{-1}$ into 15–30 μL competent cells, followed by 20 min on ice, 60–90 s heat shock at 42°C, 5 min on ice, and finally recovering the cells in 150 μL LB-Lennox medium at 37°C, 800 rpm in a plate shaker (MB100-4A) for 2 h). Then, we picked a single colony carrying test circuits from the agar plate and suspended the cells in 30 μL LB-Lennox medium with appropriate

antibiotics. 5 μL cell suspension was added into 195 μL LB-Lennox medium with appropriate antibiotics in a transparent flat-bottom 96-well plate (CytoOne).

To avoid abnormal growth at the edges of a plate, the outermost wells were not used (but were filled with 200 μL of medium). The plates were cultured at 37 °C, 1000 rpm on a plate shaker (MB100-4A) for 18 h to 23 h overnight. For induction, we added 2 μL overnight culture into 198 μL LB-Lennox medium with appropriate antibiotics and inducers on a flat clear bottom 96-well plate with black walls (Greiner Bio-one). The inducers employed in this study include N-(3-oxohexanoyl)-L-homoserine lactone, AHL (Sigma-Aldrich), aTc (CAYMAN Chemical), rhamnose (Alfa Aesar), arabinose (Acros Organics), NaAsO$_2$ (35000-1L-R, Fluka), CuSO$_4$ (SLS), ZnCl$_2$ (VWR Chemicals), PbCl$_2$ (Sigma-Aldrich) and CdCl$_2$ (Sigma-Aldrich), H$_2$O$_2$ (SLS). All stock inducer solutions have 40× concentrations, which were diluted to 1× final concentrations by adding 5 μL of stock solutions into a final volume of 200 μL culture system.

The purchased powder was used to prepare all reagents by dissolution in double distilled water or ethanol (chloramphenicol) followed by filtration through 0.22 μM syringe filters (Millipore)[16].

**Plasmid circuit construction.** We used standard molecular biology protocols in this study for circuit construction. All plasmids and their structures and sequences are listed in Supplementary Data 1. The typical genetic composition of the plasmids is illustrated in Supplementary Fig. 12. Key primers and oligonucleotides used in this study are listed in Supplementary Data 1. The dCas9 generators / dxCas9 generators, activator generators, and reporter circuits were carried by the BioBrick vector pSB4A3[35]. The sgRNA or crRNA generators were hosted on a compatible vector, p15AC separately. A third compatible vector, pSEVA221, carried the tracrRNA generators with kanamycin resistance (GenBank: JX560327). Qiaspin Miniprep Kit (Qiagen) and Monarch PCR & DNA Cleanup Kit (NEB) were used in this study for nucleic acid purification.

All the genetic parts of the dCas9 generator, dxCas9 generator, activator generator, inducible promoters, and sgRNA with aptamers at the tetraloop come from our previous published study[16]. The new versions of sgRNA were synthesized by annealing of oligonucleotides (j5 protocol[36]). PCR was used to split sgRNA to crRNA and tracrRNA, introduce mutations into gRNAs, and change the spacer sequence of crRNAs.

For designing the UASs in the experiments of hijacking small RNA/mRNA, the upstream RNA sequences immediately adjacent to the tracrRNA target region were used as the design templates for the UASs (detailed sequences are listed in Supplementary Data 1)

All the plasmids have been sequenced to confirm the sequence (Source Bioscience), except for the AGATHA DNA, which cannot be sequenced because it contains a long palindromic sequence, and thus has been checked by HpyAV restriction mapping.

**In vivo gene expression assay.** For the experiments hijacking mRNA as crRNA, after 7 h induction culture, the cultures on 96-well plate were read by a plate reader (BMG FLUOstar fluorometry with Omega Control v5.11) equipped with a 485 nm excitation laser and a 520 nm emission filter for green fluorescence measurements (Gain 700), and a 584 nm excitation laser and a 620–10 nm emission filter for red fluorescence measurements (Gain 2000). The optical density at 600 nm ($A_{600}$) was collected at the same time for calculating fluorescence/$A_{600}$ (Fluo./$A_{600}$).

For experiments hijacking small RNA OxyS from *E. coli*'s genome, the overnight cultures were diluted to 96-well plate as above, and we were first induced with aTc, rhamnose and arabinose and incubated in the plate reader at 37 °C, 700 rpm. After 3 h incubation, the cultures were induced with H$_2$O$_2$. The $A_{600}$ and fluorescence were measured every 20 min overnight, and the data shown were collected after 4 h post H$_2$O$_2$ induction.

For data with Fluorescence (a.u.) as unit in the figures except the Fig. 5, to improve accuracy, after 6 h induction culture, the cultures were fixed by adding 2 μL culture into 198 μL 1 × phosphate-buffered saline (K813-500ML, VWR) in a 96-well U-bottom plate (Thermo Fisher Scientific) with 1 mg mL$^{-1}$ kanamycin and the samples were stored at 4 °C for at least 1 h. The fixed samples on 96-well U-bottom plate were read by the Attune NxT Flow Cytometer (equipped with Attune NxT Autosampler) with Attune NxT Software v3.2.1, equipped with 488 nm excitation laser and 530/30 nm emission filter[16].

For other assays, after 6 h induction culture, the cultures on 96-well plate were read by a plate reader (BMG FLUOstar fluorometry) equipped with a 485 nm excitation laser and a 520 nm emission filter for green fluorescence measurements (Gain 700), the $A_{600}$ data was collected at the same time for calculating Fluo./$A_{600}$.

The induction conditions and target sequence employed for all the experiments in Figs. 1–5 are described as follows: For all experiments in Figs. 1, 2.5 ng mL$^{-1}$ aTc, 1.6 μM AHL and 0.08 mM arabinose were used for dCas9, crRNA, tracrRNA induction respectively. An artificial sequence LEA2 (see Supplementary Data 1) was used as the UAS and spacer. For the experiment in Fig. 2a, dCas9 was induced under P$_{tet}$ promoter with 2.5 ng mL$^{-1}$ aTc. The activator, crRNA and tracrRNA are all driven by the constitutive promoter J23106. The spacer LEA2 was employed, and a corresponding σ$^{54}$-dependent promoter with UAS LEA2 was used for reporter expression. For the experiment in Fig. 2b, artificial sequence LEA2 was employed as UAS and spacer, dCas9 was controlled by P$_{tet}$ promoter, and *rnc*, crRNA and tracrRNA expression was driven by promoters P$_{rhaB}$, P$_{lux2}$ and P$_{BAD}$

respectively. Inducer concentrations: 2.5 ng mL$^{-1}$ aTc, 1.6 μM AHL, 0.33 mM arabinose and 0.4 mM rhamnose. For the experiment in Fig. 2c, d, artificial sequence LEA2 was employed as UAS and spacer, promoters P$_{tet}$, P$_{lux2}$ and P$_{BAD}$ were used to drive the expression of dCas9, crRNA and tracrRNA respectively. The activator is driven by the constitutive promoter J23106. Inducer concentrations: 2.5 ng mL$^{-1}$ aTc, 1.6 μM AHL and 0.08 mM arabinose. For the experiment in Fig. 2e–g, details are described in the method section of library of crRNA-tracrRNA pairs with randomized hybridizing regions. For the experiment in Fig. 3b, d, e, dCas9 and activator expression were driven by the P$_{tet}$ and P$_{rhaB}$ promoters. The mRNA of RFP and the tracrRNA were transcribed from P$_{lux2}$ and P$_{BAD}$, respectively. Inducer concentrations: 2.5 ng mL$^{-1}$ aTc, 0.2 mM rhamnose, and 0.08 mM arabinose; 100 nM AHL used for mRNA expression. For the experiment in Fig. 3c, f, dCas9 and activator expression were driven by the P$_{tet}$ and P$_{rhaB}$ promoters respectively. The mRNA of RFP and the tracrRNA were transcribed from the P$_{lux2}$ and P$_{BAD}$, respectively. A gradient concentration of AHL (0.2, 0.1, 0.05, 0.03, 0.01, 0 μM) was used for mRNA expression. Inducer concentrations: 2.5 ng mL$^{-1}$ aTc, 0.2 mM rhamnose, and 0.08 mM arabinose. For the experiment in Fig. 4b, c, expression of dCas9 and activator were driven by P$_{tet}$ and P$_{rhaB}$ promoters. The tracrRNA were transcribed from P$_{BAD}$. Inducer concentrations: 2.5 ng mL$^{-1}$ aTc, 0.4 mM rhamnose, 100 nM AHL and 0.08 mM arabinose. For the experiment in Fig. 4e, expression of the dCas9 and activator were driven by P$_{tet}$ and P$_{rhaB}$ promoters. The tracrRNA were transcribed from P$_{BAD}$. Inducer concentrations: 2.5 ng mL$^{-1}$ aTc, 0.4 mM rhamnose, and 0.08 mM arabinose.

**In vitro AGATHA assay.** The in vitro AGATHA assays were performed in a 10 μL reaction volume using a previously developed in vitro transcription only cell-free system (TXO)[32]. Each reaction contained 5 ng μL$^{-1}$ DNA, 0.75 × optimized transcription/detection buffer (OTDB) (10× stock solution includes 400 mM Tris base pH 7.5 adjusted with HCl (Sigma-Aldrich), 60 mM MgCl$_2$·6H$_2$O (Sigma-Aldrich), 100 mM DTT (Melford) and 20 mM Spermidine (Alfa Aesar), 1.5 mM NTPs (Thermo Scientific™ NTP Set, Tris buffered), 1.4 units μL$^{-1}$ T7 RNA polymerase (RNAP, NEB 2 units μL$^{-1}$) RNase inhibitor (NEB), 50 nM Cas9 (NEB), 50 nM synthesized tracrRNA (IDT, Supplementary Data 1), 50 nM synthesized SARS-CoV-2 RNA fragment (IDT, Supplementary Data 1), 0.5 units μL$^{-1}$ PstI-HF (NEB) and 7.5 μM DFHBI (Tocris). SARS-CoV-2 RNA was replaced with nuclease free water (NEB) for the assay without the target RNA. For the assay with pre-incubation, everything except T7 RNAP and AGATHA DNA was mixed and first incubated at 37 °C for 10 min to allow the formation of CRISPR/Cas9 complex, then the AGATHA DNA was added and the mixture was incubated further at 37 °C for 1 h to allow the cleavage of the DNA, after which the T7 RNAP was added to the mixture to activate the Broccoli transcription. The AGATHA DNA was purified using QIAprep Spin Miniprep Kit (Qiagen) following the manufacturer's protocol.

For each assay, 35 uL reactions were made, then three replicates with 10 uL of each reaction were loaded into a black 384-well microplate with clear bottom (Greiner Bio-One). The plate was sealed with a transparent EASYseal plate sealer (Greiner Bio-One) and was incubated and measured continuously by BMG FLUOstar plate reader for 8 h at 37° C with 1 s shaking (200 rpm, double orbital) before each measurement. The settings for measuring the fluorescence were the same as for the in vivo characterization. The plate reader data was processed using Omega MARS 3.20 R2, Microsoft Excel 2013 and GraphPad Prism v9.1.1. The background of output signals was subtracted from each in vitro reaction by using its triplicate-averaged counterpart of the negative control (reporter-free) at the same time. All the data shown are mean values with standard deviation as error bars.

To prepare for cell phone imaging at the end of AGATHA assays, the microplate was placed onto the surface of a Safe Imager (S37102, Invitrogen) blue-light trans-illuminator and was covered with an amber filter in a dark environment. A mobile phone (iPhone 11) was used to acquire the images with the built-in night mode and 3 s auto exposure. No additional adjustments were made to the images.

**Calculation of fluorescence intensities.** For data collected by flow cytometer, at least 100,000 events were collected. Live bacterial cells were then gated by FCS-H and SSC-H based on a fixed interval determined from a control sample of cells with empty vector. Eventually 1,000-10,000 events were gated and analyzed for each sample. No boundaries to define 'positive' and 'negative' cell populations. The geometric mean of fluorescence of a gated population was calculated in FlowJo v10.7.1 or FlowJo 7.6.1, then corrected by the geometric mean of background fluorescence of cells with suitable empty plasmids grown in identical conditions but without inducers[16]. Corrected fluorescence values were then averaged in GraphPad Prism v9.1.1 to give the means with standard deviations (s.d.) for plotting graphs.

For data collected by the plate reader, the fluorescence values and OD values were corrected by a blank negative control (with corresponding volumes of medium and with water replacing inducers). Then, the Fluo./$A_{600}$ for each well was calculated, then corrected by that of a fluorescence-free strain. All the above steps were done in Microsoft Excel 2016. Corrected Fluo./$A_{600}$ values were then averaged in GraphPad Prism v9.1.1 to give the means with standard deviations (s.d.), which were used for plotting graphs[16].

For the dynamic range calculation, we used the calculation method described in our previous study to calculate the dynamic range and its standard deviation in Microsoft Excel 2016[16]. The negative control corrected means and standard

deviations of the data for each sample group were calculated first. We then calculated the difference between the average output value in the ON state (*Fluo (on)*) and the average output value in the OFF state (*Fluo (off)*), and the corresponding standard deviations *SD(on-off)* by the below uncertainty propagation formula:

$$SD(on - off) = \sqrt{SD(on)^2 + SD(off)^2}$$

The dynamic range *DR* and the corresponding standard deviations SD (DR) were calculated:

$$DR = (Fluo(on) - Fluo(off))/Fluo(off)$$

$$SD(DR) = \sqrt{\left(\frac{SD(on - off)}{Fluo(on) - Fluo(off)}\right)^2 + \left(\frac{SD(off)}{Fluo(off)}\right)^2} \times DR$$

For data from flow cytometry in Supplementary Fig. 2b, the standard deviations of the negative control corrected data, $SD_{cor}$, were calculated by the below uncertainty propagation formula:

$$SD_{cor} = \sqrt{SD(x)^2 + SD(nc)^2}$$

*SD(x)*: standard deviation of the mean fluorescence values of a particular sample x; *SD (nc)*, standard deviation of the mean fluorescence values of the negative control samples.

For the AGATHA sensor, the cell-free system without adding any DNA template was used as the blank sample, and the fluorescence values were corrected by the blank control.

For data shown in Supplementary Fig. 11g, we used a strain carrying empty vectors as a fluorescence-free control under each induction condition, since copper sulfate solution was observed to cause additional basal fluorescent signal. The Fluo./$A_{600}$ for each well was calculated, and then corrected by that of the fluorescence-free strain in the corresponding induction condition.

**Characterization of orthogonality and the AND logic gates.** For the characterization of orthogonality and AND logic gates (Fig. 1e, h, i), we performed high-throughput co-transformation and subsequent cell culturing steps without plating the cells on agar plates. 1.3 μL of each plasmid (miniprep of 10 mL cell culture) was added into 96-well PCR plates, and each well included a particular combination of three plasmids. 25 μL chemically competent cells of *E. coli* MC1061Δ*pspF* was added into each well on ice, and then the 96-well PCR plates were kept on ice for 20 min. A 60~90 s heat shock was performed using a PCR machine (ABI Veriti 96 well PCR Thermal Cycler), equilibrated at 42˚C, prior to the plates being placed back on ice for 5 min.

15–25 μL transformation product of each well was inoculated into a flat-bottom 96-well plate (CytoOne) with 150 μL fresh LB-Lennox medium. The plates were cultured at 37˚C, 800 rpm on a plate shaker (MB100-4A) for 2 h to recover. Then, the recovered sample of each well was diluted 4-fold by adding 50 μL culture into 150 μL LB-Lennox medium with appropriate antibiotics in another flat-bottom 96-well plate (CytoOne) for overnight culture at 37˚C, 1000 rpm on a plate shaker (MB100-4A). Then the samples were diluted 100-fold to final volume 200 μL for the second overnight culture at 37˚C, 1000 rpm. The overnight culture was diluted for 6 h induction culture followed by plate reader reading.

**Library of crRNA-tracrRNA pairs with randomized hybridizing regions.** Random sequences were generated by an online tool[37], then the sense strand and antisense strand sequences were used for the paired segments of crRNA and tracrRNA, respectively. Among them, tracrRNA retains the bulge structure and the G for the wobble base pairs. We used Excel 2016 to check whether there is an NGG in crRNA next to the spacer and exclude these sequences. Sticky ends compatible with the expression vector were designed to ensure that the DNA annealing products can be directly used for ligation.

The randomized sequences for every reprogramed crRNA and tracrRNA were ordered as complementary DNA oligonucleotides (Merck), annealed and cloned into the respective backbone plasmids. The annealing was carried out in 96-well PCR plates by mixing 2 μL of each oligonucleotide at 100 μM with 2 μL of 10x T4 DNA Ligase Reaction Buffer (NEB) and 14 μL of water. The plate was then incubated at 95 °C for 5 min in a thermocycler and allowed to cool down within the machine until reaching room temperature (c.a. 20 min). The annealed oligonucleotides produce overhangs compatible with the digested destination plasmids. The crRNA duplexes were cloned into the plasmid pLY257, previously digested with *Bbs*I-HF (NEB), and the tracrRNA duplexes were cloned into the plasmid pLY258, previously digested with *Bsa*I-HFv2 (NEB). Backbone plasmids were purified from gel after digestion using the Monarch DNA Gel Extraction Kit (NEB). Ligations were carried out in 96-well PCR plates by mixing 0.2 μL of digested plasmid (3-9 ng μL⁻¹) and 0.3 μL of annealed oligonucleotides with 0.5 μL of 2x Instant Sticky-end Ligase Master Mix (NEB) and immediately placing the plate on ice. 5 μL of chemically competent TOP10 cells were added to each well and the plate was incubated on ice for 30 min. The entire plate was heat shocked for 45 s at 42 °C in a thermocycler and incubated on ice for 2 min before adding 194 μL of LB medium to the wells.

The cells were subsequently transferred to a flat-bottom 96-well plate and incubated for 1 h at 37 °C and 1000 rpm on a plate shaker (Allsheng). After this recovery time, 10 μL of cells were spotted on LB agar containing the appropriate antibiotics on an OneWell Plate (Greiner) and incubated overnight at 37 °C. Individual colonies were picked from the plates and inoculated into flat-bottom 96-well plate containing 200 μL of LB medium, grown for 5 h at 37 °C and 1000 rpm on a plate shaker. 2 μL of cells were used to inoculate 96-deepwell plates (Starlab) containing 1.25 mL of TB medium and incubated overnight at 37 °C and 700 rpm on a plate shaker. Cells were collected by centrifuging the plates for 10 min at 1200 x *g* and the plasmids' DNA was purified using the QIAGEN Plasmid Plus 96 Miniprep Kit (Qiagen), following the manufacturers' instructions.

All plasmids were sequenced to confirm the sequence (Source Bioscience). Complementary plasmids were co-transformed into competent MC1061Δ*pspF* cells carrying the reporter plasmid pLY54 (prepared using the Mix & Go *E. coli* Transformation Kit from Zymo Research, following the manufacturer's instructions). Briefly, 2 μL of each plasmid were added to 22.5 μL of the MC1061Δ*pspF* competent cells in a 96-well PCR plate, incubated on ice for 30 min and the entire plate was heat shocked for 45 s at 42 °C in a thermocycler. After additional incubation on ice for 2 min, 175 μL of SOC medium was added to the wells, transferred to a flat-bottom 96-well plate and incubated for 1 h at 37 °C and 1000 rpm on a plate shaker. After recovery, 30 μL were transferred to a new plate containing 170 μL of SOC medium supplemented with the appropriate antibiotics (ampicillin, kanamycin and tetracycline) and incubated overnight at 37 °C and 1000 rpm on a plate shaker. Each experiment was performed on three different days, and 2 μL of the cultures were used to inoculate 198 μL of fresh LB medium supplemented with the appropriate antibiotics and grown overnight at 37 °C and 1000 rpm on a plate shaker. After overnight growth, 2 μL of cultures were used to inoculate 198 μL of fresh LB medium supplemented with the appropriate antibiotics and 2.5 ng mL⁻¹ aTc to induce dCas9 expression, on a 96-well μclear black flat-bottom plate covered with a lid (GBO) to prevent evaporation. The plate was incubated inside a FLUOstar Omega microplate reader (BMG labtech) at 37 °C and 700 rpm, for a period of 23 h. End-point fluorescence for GFP (excitation filter: 485 nm; emission filter: 520–10 nm; gain = 800) and culture optical density at 600 nm ($A_{600}$) were measured every 20 min using the Omega Control v5.11 (BMG Labtech) and data was analyzed using Omega MARS Software v3.32 (BMG Labtech). The LB medium background fluorescence and absorbance were subtracted from the readings of sample wells and Fluo./$A_{600}$ was calculated for all samples. Normalized data were produced by subtracting the negative control Fluo./$A_{600}$ from the other samples and the 6 h data was exported to GraphPad Prism v9.1.1 for plotting the figures.

The variables considered were: the average Fluo./$A_{600}$ calculated for the three biological replicates at 6 h after inoculation; the minimum free energy (MFE) of the crRNA optimal secondary structure, predicted using the RNAfold web server (http://rna.tbi.univie.ac.at/cgi-bin/RNAWebSuite/RNAfold.cgi) (Variable 1); the ΔG for crRNA and tracrRNA matching region heterodimer binding, predicted using the RNAcofold web server (http://rna.tbi.univie.ac.at/cgi-bin/RNAWebSuite/RNAcofold.cgi) (Variable 2); the homology between the crRNA, without the terminator region, and the target DNA sequence (35 nucleotides including the spacer region and the 15 nucleotides downstream) (Variable 3); the alignment scores of the crRNA and tracrRNA matching region calculated using CLUSTALW 2.1 (https://www.genome.jp/tools-bin/clustalw) (Variable 4) and the GC content of the crRNA and tracrRNA matching region (Variable 5).

The Pearson correlation analysis was conducted with the R 4.0.5 and R package "ggcorrplot", and the regression model was built with the MATLAB 2015b function "regress".

**Reporting summary.** Further information on research design is available in the Nature Research Reporting Summary linked to this article.

## Data availability

All data in the main text and the supplementary materials are provided as a Source Data file. The previously constructed plasmids pLY54(#130923), pLY76(#130963) used in this study are available from Addgene. Representative plasmids of dual-RNA mediated CRISPRa, mRNA hijackers, small RNA hijackers are available from Addgene at https://www.addgene.org/Baojun_Wang/. Source data are provided with this paper.

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

## Acknowledgements

We thank Dr Jennifer Tullet (University of Kent) for supplying the W3110 and HT115 *E. coli* strains used in this study. This work was supported by the UK Research and Innovation Future Leaders Fellowship [MR/S018875/1], Leverhulme Trust grant [RPG-2020-241], US Office of Naval Research Global grant [N62909-20-1-2036] and Wellcome Trust Institutional Translational Partnership Award. Z.X. was supported by the Natural Science Foundation of China (31771483, 61721003). J.C. was supported by the UK Global Challenges Research Fund through funding from the Engineering and Physical Sciences Research Council (EP/R01437X/1, co-funded by the UK National Institute for Health Research), as well as the UK Medical Research Council (MR/V035401/1).

## Author contributions

Y.L. and B.W. conceived the study and designed the experiments. Y.L. performed the majority of the experiments. F.P. performed the experiments related to the library of crRNA-tracrRNA pairs with randomized matching regions. X.W. and Y.L performed the experiments of the hijacking of endogenous small RNA and mRNAs. X.W. and M.L. performed the experiments for the AGATHA sensor. Y.L. F.P. and X.W. performed data analysis. S.P. and Z.X. analyzed the data from the random paired sequence library. C.F., Z.Y. and J.C. provided support and materials for the development of the in vitro RNA sensor. All authors took part in the interpretation of results and preparation of materials for the manuscript. B.W. and Y.L. wrote the manuscript with input from all co-authors. B.W. supervised and acquired the funding of the study.

## Competing interests

The authors declare no competing interests.
