## [Peer Review File · Nature Communications]

Reviewers' Comments:

Reviewer #1:

Remarks to the Author:

The authors reported the reengineering of the high programmability of crRNA-tracrRNA hybridization for *Streptococcus pyogenes* Cas9 into using mRNAs as crRNAs to trigger CRISPR function. They described the use of this strategy for in vivo mRNA sensing as well as in vitro detection of COVID RNA. While the idea itself is interesting, the level of activation is low and the generalization of the strategy to a wide range of mRNAs is missing. There are also some other issues that must be addressed.

Specific comments:

1. Fig. 1. Such a high background activation with only crRNA and dCas9 or crRNA and tracrRNA? This is not normal. The so call logic gate is really not working in this case. Please explain and fix this issue.
2. The main contribution of the paper is the potential use of full-length mRNA for activation. Unfortunately, the level of activation is about 30-fold lower than using the mRNA fragments clearly indicating the rest of the sequence impact activation. While this may be used for mRNA sensing, the sensitivity is higher questionable.
3. While the result with Ar2 is promising, the authors need to show that this can be generalized to other mRNA targets (as least four to five more in a highly predictable manner) to make this strategy attractive in general.

Reviewer #2:

Remarks to the Author:

Comments.

The authors presented the advanced RNA detection technology based on tracrRNA reprogramming for CRISPR system. In this study, the previously reported RNA capturing concept (Science, 2021, Chunlei Jiao et al.) was implemented to CRISPRa method which is optimized by various experiments including randomized crRNA-tracrRNA hybridization. Overall, the experiments are well designed and show the robustness of the new technology. The results presented support the main conclusion of the paper. I would recommend this paper for publication in nature communications after minor revisions.

Minor concerns

- Figure 1a, 3a, 4d; Please add accurate information about UAS to the GFP reporter. Also, information on exactly how UAS was designed according to the RNA to be targeted is

required in the methods section. In this regard, please describe the paragraphs (Combining the fact that ~) on page 8 of the main body in a way that is easy to understand.

- Figure 1e, The tracrRNA sequence information corresponding to WT-U5 is required.
- Optimization of the CRISPRa system: Is there a difference in GFP expression according to the distance between UAS and promoter in the GFP reporter?

Is there a GFP expression control experiment that removes the binding site of IHF?

- page 17; Please describe in detail the transformation conditions of the plasmid used in this experiment at method section. (e.g. amount of the plasmid, condition of the chemical transformation)

- Figure legend 1, 2: There is no description for the use of some abbreviations, such as UAS, dCas9, AHL and aTc.

We would like to thank the reviewers for their constructive and helpful feedback: Here we provide our point-by-point responses to their comments. The original comments were colored black, and our responses were colored blue. **Modified or inserted texts into our manuscript and supplementary information file were colored red.**

REVIEWER COMMENTS

Reviewer #1 (Remarks to the Author):

The authors reported the reengineering of the high programmability of crRNA-tracrRNA hybridization for *Streptococcus pyogenes* Cas9 into using mRNAs as crRNAs to trigger CRISPR function. They described the use of this strategy for in vivo mRNA sensing as well as in vitro detection of COVID RNA. While the idea itself is interesting, the level of activation is low and the generalization of the strategy to a wide range of mRNAs is missing. There are also some other issues that must be addressed.

We thank the comments from Reviewer #1. In this project, we achieved the programmable RNA sensing by using the CRISPR/Cas9 system. By combining it with CRISPRa, the output dynamic range of this system can reach up to 40-fold, allowing to clearly differentiate between 'Off' and 'On' states. The value of this device is demonstrated through practical application with the capability to measure changes in the concentration of target RNAs and related inducers (e.g., induction level of AHL, level of RFP gene transcripts and low concentrations of arsenic in the environment).

To further demonstrate the value and general applicability of our methods, we performed additional experiments to extend the application of this device to more endogenous RNAs, including both mRNA and small RNA. Our new data (see below) also demonstrates that using this strategy, and by hijacking of various *E. coli* endogenous RNAs, we can sense oxidative stress, as well as to detect zinc, lead, cadmium, and copper ions in the environment. We hope that the updated manuscript now demonstrates the generalization of this strategy.

Specific comments:

1# Fig. 1. Such a high background activation with only crRNA and dCas9 or crRNA and tracrRNA? This is not normal. The so call logic gate is really not working in this case. Please explain and fix this issue.

We thank Reviewer #1 for this question. Like the sgRNA-based CRISPRa system that we reported in 2019¹, this dual-gRNA-based CRISPRa system is sensitive to the low concentration of the gRNA with RNA aptamers. This only occurs in circuits which have the inducible gRNA generator.

We realized from the outset that an expression-inducible system would provide a decisive advantage for building logic gates, especially for complex tasks that require cascaded regulation. In the sgRNA case, we addressed this issue by using dxCas9 and successfully achieved CRISPR-enabled cascaded gene activation in *E. coli*¹.

In this manuscript, the optimization idea is the same as above, which is to reduce the affinity of the CRISPR complex to the gRNA (tracrRNA). We adopt two strategies and describe the results of both strategies in the original manuscript (see below, Page 3):

For the inducible system, this AND gate became less efficient due to its high sensitivity to tracrRNA leakiness from the inducible promoter input (Figure 1c). We truncated the crRNA-tracrRNA pairing region to 14 bp to weaken their pairing affinity and improved the AND gates with good

symmetry in response to the two inputs (**Figure 1g**). In addition, inspired by our previous experience with dxCas9 3.7, and how it could be used to optimize the sensitivity to the sgRNA leakiness of our CRISPRa system¹, we verified that dxCas9 can also work with our newly designed sgRNA and performed better than dCas9 (**Supplementary Figure 2**). Next, we showed that the dxCas9 could make the AND gates more symmetrical in response to the two inputs (**Supplementary Figure 3**).

We did notice that the manner in which the data was presented might result in readers overlook the paragraph above. Accordingly, we have added a new experiment to show how the truncated crRNA-tracrRNA pairing region can reduce the basal output, resulting from the tracrRNA leakiness. The data is shown in a new **Figure 1g**:

Figure 1. Dual-RNA mediated CRISPRa system reveals the programmability of crRNA-tracrRNA hybridization. (a) Schematics showing the original sgRNA mediated eukaryote-like CRISPRa system (top) and the dual-RNA mediated CRISPRa system (bottom). Green base pairs, the minimum structure required for CRISPR/Cas9 function; blue bases, RNA aptamer BoxB; red bases, the bulge structure and wobble base pairs within the crRNA-tracrRNA hybridized region. The short lines between the bases indicate the paired RNA bases. Two wobble base pairs are marked as black dots. (b) Circuit design of the crRNA-tracrRNA mediated CRISPRa system. Promoters P_{tet} , P_{lux2} , P_{BAD} drive the expression of dCas9, crRNA and tracrRNA, respectively. The activator PspFΔHTH::λN22plus is driven by constitutive promoter J23106 (Anderson promoter collection). A σ^{54} -dependent promoter with artificial upstream activating sequence (UAS) was employed to express the *sfgfp* reporter. The anhydrotetracycline (aTc), N-(3-oxohexanoyl)-L-homoserine

lactone (AHL), and arabinose are the inducers for promoters P_{tet} , P_{lux2} , P_{BAD} , respectively. **(c)** Test of the combination of three components in the crRNA-tracrRNA mediated CRISPRa, and comparison of the function of gRNAs before and after being split. Inducer concentrations: 2.5 ng mL⁻¹ aTc, 1.6 μM AHL and 0.08 mM arabinose. Combinations were achieved via presence (+) or absence (-) of inducers. The random sequence LEA2 (see in **Supplementary Table 8**) was used here as the UAS and spacer. Error bars, s.d. ($n = 4$). **(d)** Design of the library for testing reprogrammed crRNA-tracrRNA hybridized pairs. The blue letter 'N' indicates the substituted base pairs. The representation of motifs in the remaining sequences on the bottom is the same as those in **a**. **(e)** Orthogonality test of reprogrammed crRNA-tracrRNA pairs. The label 'mis' indicates the WT crRNA has a mismatched spacer (LEB3) with the target UAS (LEA2). All the other crRNAs have spacer LEA2 and a corresponding σ^{54} -dependent promoter with UAS LEA2 was used for the reporter expression. dCas9 expression was under P_{tet} promoter, and the activator expression was under a constitutive promoter J23106. sgRNA was produced under P_{lux2} promoter. The reprogrammed tracrRNAs and crRNAs were combined in pairs. Inducer concentrations: 2.5 ng mL⁻¹ aTc, 1.6 μM AHL and 0.08 mM arabinose for dCas9, crRNA and tracrRNA induction, respectively. Error bars, s.d. ($n = 3$). **(f)** Design of orthogonal AND gates based on reprogrammed crRNA-tracrRNA pairing. The diagram shows the design of the CRISPR-enabled AND gate circuit. **(g)** The AND gate function based on the original tracrRNA design and the truncated tracrRNA. Inducer concentrations: 2.5 ng mL⁻¹ aTc, 1.6 μM AHL, and 0.08 mM arabinose for dCas9, crRNA, tracrRNA induction respectively. Normalization was performed with the output of WT tracrRNA-based device at 'On' state. Error bars, s.d. ($n = 3$). **(h)** The heat map on the right shows the outcome of an orthogonality test of the WT and 4 randomly generated crRNA-tracrRNA paired sequences. Inducer concentrations: 2.5 ng mL⁻¹ aTc, 1.6 μM AHL, and 0.08 mM arabinose for dCas9, crRNA, tracrRNA induction respectively. Error bars, s.d. ($n = 3$). **(i)** Each orthogonal crRNA-tracrRNA mediated CRISPRa device displays the Boolean AND logic profile. The cartoon above shows the presence or absence of the components of the CRISPR complex under different induction conditions. The data in the active state and induction conditions are the same as in **g**. Error bars, s.d. ($n = 3$). a.u., arbitrary units.

At the same time, we revised the last paragraph of Section 1 of Results to emphasize that the circuits that we used for the creation of the heatmaps (new **Figure 1h, i**) have a truncated crRNA-tracrRNA pairing region:

We randomly generated four 14 bp sequences for the crRNA-tracrRNA matching region and tested their orthogonality and functionality within the AND gates. Accordingly, we showed that orthogonality could be achieved following the principle of complementary base pairing, and that all circuits retain their function as AND gates (**Figure 1h, i**).

We believe that whether the basal output level of an AND gate device is acceptable depends upon the specific application scenario. This requirement may become critical for complex cellular computing or cascaded regulation (for simple logic circuits, this pre-requisite can be relaxed). We have provided more than one optimization methods and demonstrated the effectiveness and feasibility of these strategies. In future, others will be able to continue to optimize the system according to the principles proposed, depending upon their needs and their particular application.

#2 The main contribution of the paper is the potential use of full-length mRNA for activation. Unfortunately, the level of activation is about 30-fold lower than using the mRNA fragments clearly indicating the rest of the sequence impact activation. While this may be used for mRNA sensing, the sensitivity is higher questionable.

Thank you for raising this point. In our manuscript, **Figure 3d** clearly reflects the output differences between using the mRNA fragments and the full-length mRNA. However, by truncating the tracrRNA, we can improve the output level and dynamic range significantly. Further, in **Figures 3f** and **3g**, we show that the output level can be further improved through a positive feedback design and achieve an increased dynamic range, up to 50-fold.

Figure 3. (d) 5' end truncation on tracrRNA improved mRNA-mediated CRISPR function. The cartoon shows the test of component combinations: mRNA fragment or whole mRNA as crRNA with 5'-end extended tracrRNA or 5'-end truncated tracrRNA (s-tracrRNA). The mRNA fragment used has the same sequences as that in the mRNA target site. R2 used as controls. For the combination of mRNA and tracrRNA-R2, the same data were used as the data in **b**. Inductions are the same as in **b**. Statistical difference was determined by a two-tailed Welch's *t* test: mRNA + s-tracrRNA-R2, $p = 0.0005$, $t = 43.63$. Error bars, s.d. ($n = 3$) **(e)** The same sequence from mRNA fragment and whole mRNA resulted in altered CRISPR functions. The raw data obtained from R2 site are the same as that in **d**, and inductions are the same as in **b** and **d**. In each group, the data are normalized by the CRISPRa output level of the sample with R3 site.

Figure 3. (f) Diagram shows the configuration of the original reporter circuit and the version with a positive feedback loop (PFB) added, with results shown on the right. The input components were controlled by the same promoters used in **b**. A gradient concentration of AHL (0.2, 0.1, 0.05, 0.03, 0.01, 0 μM) was used for mRNA expression. Inducer concentrations: 2.5 ng mL^{-1} aTc, 0.2 mM rhamnose, and 0.08 mM arabinose. **(g)** Dynamic range calculated from **f**. Error bars, s.d. ($n = 3$). a.u., arbitrary units.

Although we have shown how to optimize the output level of CRISPRa system using a full-length mRNA (as crRNA), comparable to that of using the fragmented mRNAs (as crRNA), we would like to point out again that absolute levels or dynamic ranges of the outputs should be contextualized regarding specific application scenarios.

When we discuss the sensitivity required to detect mRNA *in vivo*, we need to have a specific engineering goal. For example, in **Figure 3f**, the mRNA level induced by 13 nM AHL was sufficient to be detected, which is at a low induction level for the P_{lux2} promoter. We would argue that this is an unexpectedly (high) sensitivity.

In a second example, in **Figure 4**, mRNA transcription can be induced by 0.31 μM NaAsO_2 , indicating a sensitivity comparable to the state of the art for artificially designed arsenic biosensors². For the AGATHA system (**Figure 5**), because RNA sensors usually enable inclusion of an isothermal amplification step *in vitro*, as a programmable reporter system, the concentration of the synthetic RNAs that we used is sufficient to demonstrate the functionality and potential application of our sensor.

Please note that we discussed in detail in the manuscript that the context of mRNA target site does influence the CRISPRa activity (**Figure 3e**), and we indicated that such effect is difficult to predict at this early stage (see **Discussion**, Page 17).

#3 While the result with Ar2 is promising, the authors need to show that this can be generalized to other mRNA targets (as least four to five more in a highly predictable manner) to make this strategy attractive in general.

We thank the reviewer for the suggestion. In order to further demonstrate the generalization of our methods, we have applied this system to build four more *E. coli* endogenous RNA biosensors using the same strategy. By screening the RNA sites of the four sensors, we have identified many functional target sites. The most efficient RNA target site for each sensor was selected for the subsequent hijacking of the endogenous mRNA and small RNA. The results showed that, by programmatically and systematically hijacking of endogenous mRNA/small RNA *in vivo* and simply changing the UAS and tracrRNA, we can use the CRISPRa system to sense the presence of hydrogen peroxide, zinc, lead, cadmium and copper ions in the environment (see below, Page 13):

We subsequently generalized this same strategy to other environment-responsive genes in the *E. coli* genome, including the oxidative stress-related small RNA OxyS, the mRNA of zinc/lead responsive gene *zraP*, the mRNA of zinc/cadmium responsive gene *zntA*, and the mRNA of copper responsive gene cluster *cusCFAB* were chosen as the targets^{2,3}.

To do this, we first isolated the four transcriptional units from the *E. coli* genome and inserted them into the AHL-induced circuits. We selected 5 to 9 target sites without any sequence restriction for each target RNA, and then screened the sites using the corresponding tracrRNA generators (with aforementioned optimized design), reporter circuits and AHL induction (**Supplementary Figure 9**). By picking the most efficient site on each RNA for further testing the specificity of CRISPRa, we were able to show that the matched tracrRNA is necessary for the CRISPRa function (**Supplementary Figure 10**).

Finally, we then demonstrated that the hijacking of endogenous small RNA OxyS and mRNA of *zraP*, *zntA*, or *cusCFAB* can also be used to trigger CRISPRa for sensing hydrogen peroxide, zinc, lead, cadmium and copper ions, respectively. Among them, we demonstrated that the CRISPRa sensors based on small RNA OxyS, *zraP* mRNA and *cusCFAB* mRNA can detect the presence of hydrogen peroxide, zinc, lead, or copper ions with a high dynamic range. However, the sensor based on *zntA* mRNA has a relatively low dynamic range in response to zinc, indicating the need of further optimization in some specific cases (**Supplementary Figure 11**).

Overall, we conclude that endogenous small RNA and mRNAs can be hijacked by Cas9, and used to monitor genomic transcriptional activity and connect the cellular gene regulatory network to an artificial actuating or reporting gene circuit.

Supplementary Figure 9. Hijacking of mRNAs cloned from the *E. coli* genome as crRNAs. (a) Schematic shows the structures of the four environment-responsive genes cloned from the *E. coli* genome. All the genes were cloned and placed downstream of a P_{Lux2} promoter. The red line and the corresponding number indicate the tracrRNA binding sites we tested and its distance from the predicted transcription start site. The asterisk indicates that the promoter contains a mutation that was accidentally introduced during construction, which may affect the quantitative relationship of the function of this circuit to the control circuit but does not affect the site screening. **(b)** The CRISPRa output from all tested target sites on the small RNA *OxyS* and downstream region suspected of being transcribed. The data are normalized with a positive control data from the optimized mRNA hijacking device of the arsenic sensor (described in **Figure 5**). The 'On' indicates that transcription of mRNA was induced using $0.1 \mu\text{M}$ AHL, and the 'Off' state indicates that no AHL induction was used. The dCas9, activator and tracrRNA were always induced by 2.5 ng mL^{-1} aTc, 0.4 mM rhamnose, and 0.08 mM arabinose, respectively. The double asterisk indicates that

the promoter P_{pspA} contains a mutation accidentally introduced during construction, which is not expected to affect the purpose of the experiment. Error bars, s.d. ($n = 3$). **(c)** The CRISPRa output from all tested target sites on the mRNA of *zraP*. The experimental conditions and data processing method are the same as in **b**. The asterisk indicates that the reporter gene *sfgfp* contains a mutation (K140N) that was accidentally introduced during construction. Error bars, s.d. ($n = 3$). **(d)** The CRISPRa output from all the tested target sites on the mRNA of *zntA*. The experimental conditions and data processing method are the same as in **b**. Error bars, s.d. ($n = 3$). **(e)** The CRISPRa output from all tested target sites on the mRNA of *cusCFAB*. The experimental conditions and data processing method are the same as in **b**. The asterisk indicates that the reporter gene *sfgfp* contains a mutation (N146S) that was accidentally introduced during construction, and the double asterisk indicates that the promoter P_{pspA} contains a mutation accidentally introduced during construction. Both mutations are not expected to affect the purpose of the experiment. Error bars, s.d. ($n = 3$).

Supplementary Figure 10. Specificity test of the four candidate mRNA hijacking devices. **(a)** The output from the CRISPRa system triggered by small RNA OxyS and the tracrRNA targeting site *oxyS-86*. A wild type (WT) tracrRNA was employed here as a mis-matched tracrRNA as the control. The 'On' indicates that transcription of mRNA was induced by 0.1 μM AHL, and the 'Off' state indicates that no AHL induction was used. The dCas9, activator and tracrRNA were always induced by 2.5 ng mL⁻¹ aTc, 0.4 mM rhamnose, and 0.08 mM arabinose, respectively. Error bars, s.d. ($n = 3$). **(b)** The output from the CRISPRa system triggered by *zraP* mRNA and the tracrRNA targeting site *zraP-22*. The experimental conditions are the same as in **a**. Error bars, s.d. ($n = 3$). **(c)** The output from the CRISPRa system triggered by *zntA* mRNA and the tracrRNA targeting site

zntA-1034. The experimental conditions are the same as in **a**. The P_{lux2} promoter for *zntA* mRNA contains a mutation that was accidentally introduced during construction, which is not expected to affect the purpose of the experiment. Error bars, s.d. ($n = 3$). **(d)** The output from the CRISPRa system triggered by *cusCFAB* mRNA and the tracrRNA targeting site *cusCFAB*-656. The experimental conditions are the same as in **a**. Error bars, s.d. ($n = 3$).

Supplementary Figure 11. Hijacking of endogenous RNAs as crRNAs to detect environmental stress induced transcription. **(a)** Schematic showing the mechanism of hijacking the endogenous mRNA of the environment-responsive operons of *E. coli* to activate reporter expression by CRISPRa. **(b)** The tracrRNA targeting RNA site *oxyS*-86 and corresponding promoter were employed to hijack of small RNA OxyS in *E. coli*. A wild type tracrRNA was used as

the negative control. Expression of the dCas9 and activator were driven by P_{tet} and P_{rhaB} promoters respectively. The tracrRNA was transcribed from P_{BAD} . Inducer concentrations: 2.5 ng mL⁻¹ aTc, 0.4 mM rhamnose, and 0.08 mM arabinose. A gradient concentration of H₂O₂ (0.000, 0.008, 0.016, 0.031, 0.063, 0.125, 0.250, 0.500, 1.000 mM) was used for inducing transcription of the small RNA OxyS. Error bars, s.d. ($n = 3$). **(c)** The tracrRNA targeting RNA site *zraP*-22 and corresponding promoter were employed to hijack of mRNA of *zraP* gene in *E. coli*. A wild type tracrRNA was used as the negative control. Expression of the dCas9 tracrRNA, and activator were driven by the same promoters and induction conditions as in **b**. A gradient concentration of ZnCl₂ (0.000, 0.008, 0.016, 0.031, 0.063, 0.125, 0.250, 0.500 mM) was used for inducing transcription of the mRNA of *zraP*. Error bars, s.d. ($n = 3$). **(d)** For the same system described in **c**, a gradient of PbCl₂ (0.000, 0.008, 0.016, 0.031, 0.063, 0.125, 0.250, 0.500, 0.870 mM) was used for inducing transcription of the mRNA of *zraP*. Error bars, s.d. ($n = 3$). **(e)** The tracrRNA targeting RNA site *zntA*-1034 and corresponding promoter were employed to hijack of mRNA of *zntA* gene in *E. coli*. A wild type tracrRNA was used as the negative control. Expression of the dCas9 tracrRNA, and activator were driven by the same promoters and induction conditions as in **b**. A gradient concentration of CdCl₂ (0.000, 0.004, 0.008, 0.016, 0.031, 0.063, 0.125, 0.250 mM) was used for inducing transcription of the mRNA of *zntA*. Error bars, s.d. ($n = 3$). **(f)** For the same system described in **e**, A gradient concentration of ZnCl₂ (0.000, 0.008, 0.016, 0.031, 0.063, 0.125, 0.250, 0.500 mM) was used for inducing transcription of the mRNA of *zntA*. Error bars, s.d. ($n = 3$). **(g)** The empty vector (\emptyset) was used to replace WT tracrRNA generator as the negative control to reduce the interference by the accumulation of leaky expressed sfGFP. For the data chart on the left, the condition is the same as in **e** except an empty vector (\emptyset) was used to replace the tracrRNA generator as the negative control. For the data chart in the middle, the condition is the same as in **f** except an empty vector was used in the negative control. For the data chart on the right, the tracrRNA targeting RNA site *cusCFAB*-656 and corresponding promoter were employed to hijack of the mRNA of *cusCFAB* gene cluster in *E. coli*. Expression of the dCas9 tracrRNA, and activator were driven by the same promoters and induction conditions as in **b**. A gradient concentration of CuSO₄ (0.000, 0.016, 0.031, 0.063, 0.125, 0.250, 0.500, 1.000, 2.000 mM) was used for inducing transcription of the mRNA of *cusCFAB*. Since the binding of copper ions to biomacromolecules in cells causes changes in basal fluorescence level, the control and data calculation methods for this set of experiments are different from others (see in **Methods**). Error bars, s.d. ($n = 3$).

In all of the above experiments, we demonstrate that the CRISPRa hijacking of the *zntA* mRNA gives a weak functional output (but still has detectable fluorescence increase), while the hijacking of the small RNA OxyS, mRNA of *zraP* and mRNA of *cusCFAB* are all sufficient to achieve a considerable change in the output dynamic range. Together with other examples already included in our original manuscript (mRNA of RFP, mRNA of *arsRBC*, and the genomic RNA of SARS-CoV-2), in total, we now provide 7 examples investigated involving small RNA, mRNA, viral RNA – all demonstrating the broad functionality that can be achieved using our strategy.

As we have discussed in the original manuscript, we found that the mRNA context and the sequence preference of the Cas effector protein make it difficult to predict the optimal target site prior to screening. For sequence preference, as we show in **Figures 2e, f, g**, we can build a random sequence library of crRNA-tracrRNA matching region to reveal the complexity of this challenge. Regarding the potential impact of the context sequence on mRNA detection, we have illustrated this clearly in **Figure 3 and Figure 4**.

Considering the complexity of predicting the optimal target site as previously mentioned, we propose that screening available functional sites is a feasible and efficient strategy. In all of the seven cases that we investigated, screening 3 – 9 sites is sufficient to discover the functional site for further engineering, which is a readily attainable and appropriate screening scale in any academic or industrial application. We hope that all these results now clearly demonstrate the generality of our strategy for detecting various mRNA targets.

Reviewer #2 (Remarks to the Author):

Comments.

The authors presented the advanced RNA detection technology based on tracrRNA reprogramming for CRISPR system. In this study, the previously reported RNA capturing concept (Science, 2021, Chunlei Jiao et al.) was implemented to CRISPRa method which is optimized by various experiments including randomized crRNA-tracrRNA hybridization. Overall, the experiments are well designed and show the robustness of the new technology. The results presented support the main conclusion of the paper. I would recommend this paper for publication in nature communications after minor revisions.

We thank Reviewer #2 for the positive comments. It is worth emphasizing that this work was independently conducted without knowing/building on the study published in April 2021 (Science, 2021, Chunlei Jiao et al.). Following the publication of this paper, we released a preprint of our work on *bioRxiv* (doi: <https://doi.org/10.1101/2021.05.24.445356>, May 2021). The similarity between our research topics and some conclusions represented a challenge although we do now clearly differentiate our work. As always, we welcome the publication and reporting mutually supporting discoveries.

Minor concerns

#1 - Figure 1a, 3a, 4d; Please add accurate information about UAS to the GFP reporter. Also, information on exactly how UAS was designed according to the RNA to be targeted is required in the methods section. In this regard, please describe the paragraphs (Combining the fact that ~) on page 8 of the main body in a way that is easy to understand.

We thank the reviewer for the suggestions. We now provide accurate information about UAS design for Figures 1a, 3a, 4d by revising the corresponding texts in **Figure 1** legend as shown below:

A random sequence LEA2 (see **Supplementary Tables 8, 9**) was used here as the UAS and spacer.

And revised the **Figure 3** legend:

The upstream RNA sequences immediately adjacent to the tracrRNA target sites were used as the design templates for the UASs (see **Supplementary Table 9**).

And revised **Figure 4** legend:

The upstream RNA sequence immediately adjacent to the tracrRNA target site was used as the design template for the UAS for CRISPRa activation (see **Supplementary Table 9**).

We also updated the Methods: (see below, Page 20):

For designing the UASs in the experiments of hijacking small RNA/mRNAs, the upstream RNA sequences immediately adjacent to the tracrRNA target region were used as the design templates for the UASs (detailed sequences are listed in **Supplementary Table 9**)

To facilitate the reader's understanding, we revised the mentioned paragraph on Page 8 as:

By combining design constraints around the programmability of the spacer sequence and its downstream repeat region, we can infer that the sequence of the whole crRNA can be altered without destroying the function of the CRISPR/Cas9 system. Subsequently, we deduced that any RNA sequence may become a crRNA through dual recognition by a cognate programmed tracrRNA and a matched target DNA.

#2- Figure 1e, The tracrRNA sequence information corresponding to WT-U5 is required.

We thank the reviewer for this helpful suggestion which helps readers to better understand the design of the experiment. Accordingly, we have updated the **Figure 1** (see below) as suggested to include such information:

#3 - Optimization of the CRISPRa system: Is there a difference in GFP expression according to the distance between **UAS** and **promoter** in the GFP reporter?

This is an excellent question as we know that the output level can be affected by the distance between UAS and the core promoter region. We also note that the double helix structure of DNA also affects the function of CRISPRa by affecting the spatial orientation of the activator (see below figure).

Figure extracted from Yang Liu *et al.* (2019, *Nature Commun.*)

The principles behind such variation have been reported in our previous published articles^{1,4}. In this study, we now use a uniform UAS-to-promoter distance in order to rule out the potential effects of this variable on the experimental results.

#4 Is there a GFP expression control experiment that removes the binding site of IHF?

We thank the reviewer for the question. As we mainly focus on using this existing CRISPRa system to study the programmability of crRNA and tracrRNA pairing, rather than the mechanism of

the CRISPRa system itself, this is not the case. However, it is worthy of note that the necessity of IHF was investigated in our previous study on CRISPRa device¹, although this is not discussed in our previous paper.

As reported in previous studies, IHF binding site is not required for all σ^{54} -dependent promoters, whereas the flexibility of the DNA loop determines the necessity of IHF binding site for a σ^{54} -dependent promoter. For example, promoter *glnAp2* has been reported as a σ^{54} -dependent promoter without IHF binding site⁵. In our study published in 2019, we used the DNA loop fragment of *glnAp2* promoter to make hybrid σ^{54} -dependent promoters used in CRISPRa. Some of these hybrid promoters showed activity (see Figure below), which indicates that IHF binding site is not a necessity for the σ^{54} -dependent promoters used in our CRISPRa in some cases¹.

Figure from Yang Liu *et al.* (2019, *Nature Commun.*)

#5 - page 17; Please describe in detail the transformation conditions of the plasmid used in this experiment at method section. (e.g. amount of the plasmid, condition of the chemical transformation)

We thank the reviewer for the suggestion. Accordingly, we now provide the detailed protocol of transformation by revising the **METHODS** as below (see Page 19):

We co-transformed the desired circuits into chemically competent *E. coli* MC1061 Δ *pspF* cells using a typical chemical transformation protocol (i.e. adding 1 μ L of each plasmid DNA with concentration from 10-100 ng μ L⁻¹ into 15 – 30 μ L competent cells, followed by 20 min on ice, 60 – 90 s heat shock at 42°C, 5 min on ice, and finally recovering the cells in 150 μ L LB-Lennox medium at 37 °C, 800 rpm on a plate shaker (MB100-4A) for 2h).

#6 - Figure legend 1, 2: There is no description for the use of some abbreviations, such as UAS, dCas9, AHL and aTc.

We thank the reviewer for the suggestion. We revised the main text where the dCas9 now appears first as below (see Page 3):

An experiment permuting three induction conditions indicated that only when crRNA, tracrRNA and nuclease-dead Cas9 (dCas9) were all induced did the CRISPRa system give the highest output (**Figure 1b, c**).

And we revised the legend of **Figure 1b** (where the abbreviations UAS, AHL and aTc appear first) as below (see Page 5, accordingly):

A σ^{54} -dependent promoter with artificial upstream activating sequence (UAS) was employed to express the *sfgfp* reporter. The anhydrotetracycline (aTc), N-(3-oxohexanoyl)-L-homoserine lactone (AHL), and arabinose are the inducers for promoters P_{tet} , P_{lux2} , P_{BAD} , respectively.

References

- 1 Liu, Y., Wan, X. & Wang, B. Engineered CRISPRa enables programmable eukaryote-like gene activation in bacteria. *Nat. Commun.* **10**:3693, (2019).
- 2 Wang, B. J., Barahona, M. & Buck, M. A modular cell-based biosensor using engineered genetic logic circuits to detect and integrate multiple environmental signals. *Biosensors & Bioelectronics* **40**, 368-376, (2013).
- 3 Altuvia, S., WeinsteinFischer, D., Zhang, A. X., Postow, L. & Storz, G. A small, stable RNA induced by oxidative stress: Role as a pleiotropic regulator and antimutator. *Cell* **90**, 43-53, (1997).
- 4 Liu, Y. & Wang, B. A novel eukaryote-like CRISPR activation tool in bacteria: features and capabilities. *Bioessays* **42**:e1900252, (2020).
- 5 Huo, Y. X. *et al.* IHF-binding sites inhibit DNA loop formation and transcription initiation. *Nucleic Acids Res.* **37**, 3878-3886, (2009).

Reviewers' Comments:

Reviewer #1:

Remarks to the Author:

The authors have adequately addressed all my concerns with new experiments. I am happy with the revision.

Reviewer #2:

Remarks to the Author:

The authors adequately addressed all of my concerns.